# A Review of Photoelectrocatalytic Reactors for Water and Wastewater Treatment

Stuart McMichael *, Pilar Fernández-Ibáñez [ID] and John Anthony Byrne *[ID]

Nanotechnology and Integrated Bioengineering Centre, Ulster University, Newtownabbey BT37 0QB, UK; p.fernandez@ulster.ac.uk
* Correspondence: mcmichael-s@ulster.ac.uk (S.M.); j.byrne@ulster.ac.uk (J.A.B.); Tel.: +44-28-9036-8941 (J.A.B.)

**Abstract:** The photoexcitation of suitable semiconducting materials in aqueous environments can lead to the production of reactive oxygen species (ROS). ROS can inactivate microorganisms and degrade a range of chemical compounds. In the case of heterogeneous photocatalysis, semiconducting materials may suffer from fast recombination of electron–hole pairs and require post-treatment to separate the photocatalyst when a suspension system is used. To reduce recombination and improve the rate of degradation, an externally applied electrical bias can be used where the semiconducting material is immobilised onto an electrically conducive support and connected to a counter electrode. These electrochemically assisted photocatalytic systems have been termed "*photoelectrocatalytic*" (PEC). This review will explain the fundamental mechanism of PECs, photoelectrodes, the different types of PEC reactors reported in the literature, the (photo)electrodes used, the contaminants degraded, the key findings and prospects in the research area.

**Keywords:** photoelectrocatalytic; electrochemical assisted photocatalysis; PEC reactor; water; wastewater; photoelectrodes; advanced oxidation process



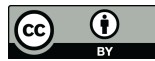

## 1. Introduction

The current water crisis affects billions of people globally due to the lack of access to a safe source of drinking water (2 billion people), water scarcity (1.9 billion people) and water storages (2.55 billion people) [1,2]. Coupled with the world's increasing population and wealth, there will be an increasing demand for water industrially, agriculturally and domestically, leading to further increases in freshwater withdrawals and, as the effects of climate change alter our environment, more countries will become water stressed [3]. Consequently, now and in the coming years, how we use and treat water, greywater and wastewater will become more important. Furthermore, there is the added complexity of removing priority substances and contaminants of emerging concern (organic micropollutants) which urban wastewater treatment plants are not designed for [4]. Therefore, there is the need for new water treatment technologies, whether for isolated communities, treating greywater before secondary use or wastewater (municipal or industrial) treatment before being discharged into water bodies. A suitably designed photoelectrocatalytic (PEC) reactor is one potential solution, and if necessary, this can be combined with other technologies to produce a multi-step water treatment system. The term "*photoelectrocatalysis*" is stated in the IUPAC Recommendations 2011 as "*electrochemically assisted photocatalysis. The role of the photocatalyst is played by a photoelectrode, often a semiconductor*" [5]. Therefore, before examining PEC reactors, it is beneficial to first understand the principles of photocatalysis.

## 2. Photocatalysis

Photocatalysis is the "*change in the rate of a chemical reaction or its initiation under the action of ultraviolet, visible or infrared radiation in the presence of a substance—the photocatalyst—that absorbs light and is involved in the chemical transformation of the reaction partners*"—IUPAC Gold Book [6]. Photocatalysis has been used to inactivate microorganisms [7], reduce metal

ions to less toxic oxidation states [8] and degrade chemical compounds, for example, dyes, pharmaceuticals, pesticides and phthalates [9,10]. This occurs because the photocatalyst is "*able to produce, upon absorption of light, chemical transformations of the reaction partners. The excited state of the photocatalyst repeatedly interacts with the reaction partners forming reaction intermediates and regenerates itself after each cycle of such interactions*"—IUPAC Gold Book [6]. The primary reactions are reduction and oxidation (redox) at the surface of the photocatalyst, which degrades the contaminate directly or indirectly by the generation of reactive oxygen species (ROS) (Figure 1).

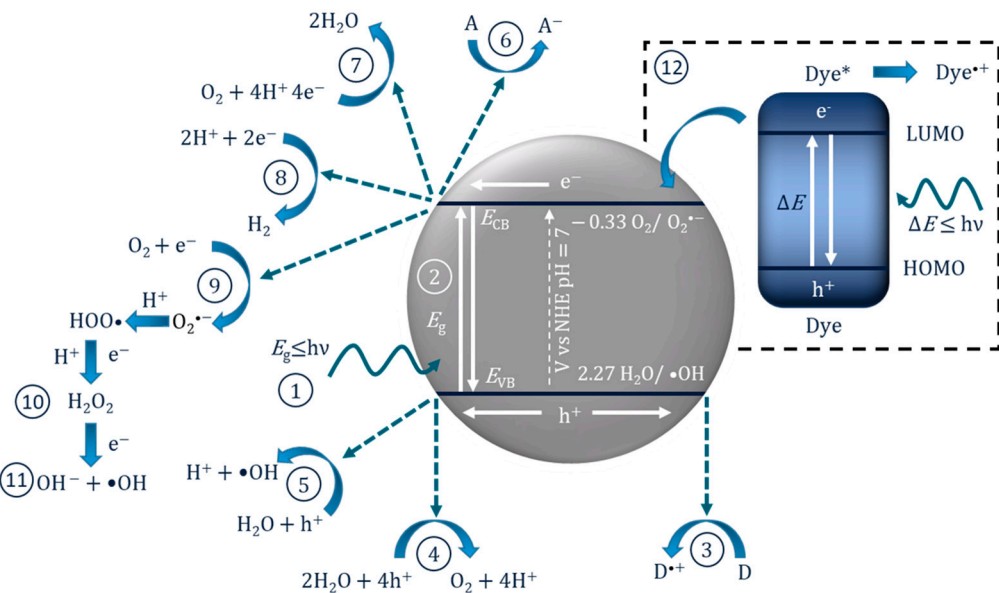

**Figure 1.** Diagram of the photocatalytic mechanism on a semiconducting material. (1) Photon absorption, (2) photoexcitation and recombination, (3) donor electron transfer, (4) oxygen evolution reaction, (5) oxidation of water to form hydroxyl radical, (6) electron transfer to an electron acceptor, (7) oxygen and proton reduction to water, (8) proton reduction to hydrogen, (9) oxygen reduction to superoxide, (10) formation of hydrogen peroxide (11) formation of hydroxyl radical, (12) dye sensitisation (* excited state) and electron transfer to the conduction band.

In the first step (1) the semiconductor absorbs an incident photon that has an energy greater than the bandgap of the material and, provided recombination does not occur, the photogenerated electron ($e^-$) and hole ($h^+$) can migrate to the surface of the photocatalyst, which will result in oxidation (3–5) and reduction (6–9) reactions [11]. For an oxidation reaction to take place, the valence band edge potential ($E_{VB}$) has to be more positive than the required reaction potential; for example, the hydroxyl radical ($\bullet$OH (5)) has a reduction potential of +2.29 V (NHE at pH 7), and in the instance of $TiO_2$, the $E_{VB}$ is +2.69 V (NHE at pH 7), and, therefore, is thermodynamically capable of oxidising water to produce the $\bullet$OH [12]. For the reduction reactions, the conduction band edge potential ($E_{CB}$) has to be more negative than the required reaction reduction potential; for example, the superoxide radical anion ($O_2^{\bullet-}$) has a reduction potential of −0.33 V (NHE at pH 7), and in the instance of $TiO_2$ (anatase), the $E_{CB}$ is −0.51 V (NHE at pH 7), and, therefore, is thermodynamically capable of reducing $O_2$ to $O_2^{\bullet-}$ [13]. Other common photocatalytic oxidation and reduction reactions with the corresponding reduction potentials are given in Table 1.

**Table 1.** Reduction and oxidation reactions and potentials from [14].

| Reaction | Potential/V vs. NHE at pH 7 | Application |
|---|---|---|
| $OH^- + h^+ \rightarrow \bullet OH$ | +2.29 | ROS generation |
| $H_2O + 4h^+ \rightarrow O_2 + 2H^+$ | +0.82 | Water splitting |
| $O_2 + 2e^- + 2H^+ \rightarrow H_2O_2$ | +0.281 | ROS generation |
| $O_2 + e^- + H^+ \rightarrow HO_2^\bullet$ | −0.05 | ROS generation |
| $O_2 + e^- \rightarrow O_2^{\bullet -}$ | −0.33 | ROS generation |
| $2H^+ + 2e^- \rightarrow H_2$ | −0.41 | Water splitting |
| $CO_2 + 2e^- + 2H^+ \rightarrow CO + H_2O$ | −0.53 | $CO_2$ reduction |

The removal of dyes from wastewater is challenging and of major environmental concern as an estimated 700,000 tonnes are produced yearly and after use are discarded into water bodies, often without any post-treatment [15]. Consequently, testing for the degradation of dyes and other photosensitive compounds has been conducted [16–19]. This is partly because it is a simple and low-cost method of testing new photocatalytic materials. However, if the excitation wavelengths of the irradiation source overlap with the absorption spectrum of the dye, a different mechanism can take place (Figure 1(12)), where the dye can become photoexcited and inject an electron into the conduction band of the semiconductor (SC), and, in the process, the dye is oxidised [20]. The injected electron in the SC can then result in a reductive reaction if the conduction band edge has a suitable potential and can result in the formation of $O_2^{\bullet -}$, which can lead to further degradation of the dye. These reactions can be expressed as Equations (1)–(5).

$$Dye + h\nu \rightarrow Dye* \tag{1}$$

$$Dye* + SC \rightarrow Dye^{\bullet +} + SC\left(e_{CB}^-\right) \tag{2}$$

$$SC\left(e_{CB}^-\right) + O_2 \rightarrow O_2^{\bullet -} + SC \tag{3}$$

$$Dye^{\bullet +} + OH^- \rightarrow Dye + \bullet OH \tag{4}$$

$$Dye^{\bullet +} \rightarrow \text{degradation products} \tag{5}$$

In these instances, it may be inappropriate to call it a photocatalytic reaction as the molecule/dye is the photoexcited species, so it should instead be correctly reported as a "*catalysed photoreaction*", as the semiconductor may only be acting as a catalyst [21]. This has been demonstrated using an $InVO_4/BiVO_4$ heterostructured semiconductor for the degradation of methylene blue. The results showed improved degradation compared to only UV–Vis (i.e., no catalyst) when using light >560 nm, which can excite methylene blue but is below the bandgap of the material and, therefore, the semiconductor was only acting as a catalyst for the photoreaction [22]. Accordingly, when testing for the removal of dyes and photosensitive compounds, it is critical to analyse the true mechanism for degradation so as to not overstate the effectiveness of a photocatalyst.

When the photocatalyst is used as a powder suspension for water treatment, post-processing is required to remove the photocatalyst from the treated solution. While possible, this adds extra cost and complexity to the system. To avoid this problem, immobilising the photocatalyst onto a substrate has been studied [23]; however, immobilisation reduces the specific surface area in contact with the solution and introduces a mass transport problem to the system, meaning immobilised systems are typically less effective than suspension systems [24]. Photocatalytic materials may also exhibit short electron–hole lifetimes (i.e., fast recombination) and low mobility, in which the diffusion length of the electron is normally larger than the diffusion length of the hole [25]. Therefore, the application of an external electrical bias in PECs can be used to overcome these problems.

## 3. Photoelectrocatalysis

In PECs, the photocatalyst or semiconducting materials are commonly used to make a photoelectrode, which is defined by IUPAC Recommendations 2011 as an *"electrode capable of initiating electrochemical transformations after absorbing ultraviolet, visible, or infrared radiation"* [5]. The photoelectrode is generally produced by immobilising a semiconducting material onto an electrically conducting supporting substrate. Examples of the latter include metals [26], carbonous materials [27] or conductive films such as transparent indium-doped tin oxide (ITO) or fluorine-doped tin oxide (FTO). The conductive film is ordinarily deposited on borosilicate glass or quartz; of the two films, FTO has lower resistivity after annealing, higher temperature stability, lower cost and has displayed improved photocurrents [28].

In PECs, the photoanode is normally an n-type semiconductor (electrons are the majority carriers) as these materials exhibit anodic photocurrents (Figure 2a), while p-type materials (holes are the majority carriers) display cathodic photocurrents (Figure 2b) and therefore are used as photocathodes; thus, a photocurrent is only observed when the minority charges are involved [29]. Photocathodes (p-type) have had less attention in the literature as compared to photoanodes due to instability in aqueous environments under irradiation, and due to self-reduction, e.g., $Cu^{2+}$ to Cu [30] or self-oxidation, e.g., $Cu_2O$ to CuO [31].

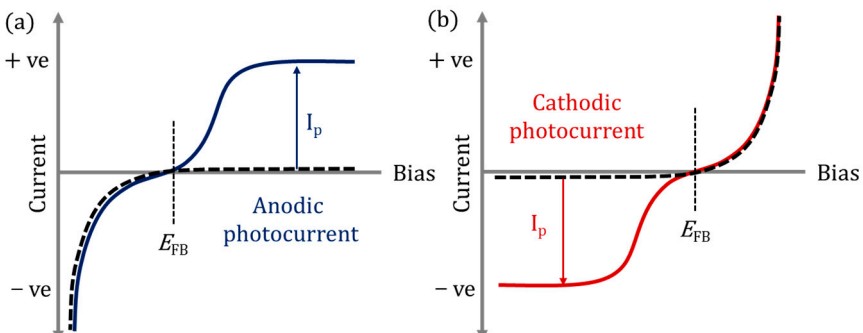

**Figure 2.** (**a**) Anodic photocurrent ($I_p$) response, (**b**) cathodic photocurrent ($I_P$) response; solid lines under irradiation; dashed lines in dark; $E_{FB}$ flat band potential.

The PEC mechanism in Figure 3 demonstrates the process when using an n-type photoanode with an inert non-semiconducting counter electrode with an externally applied bias. The mechanism on the photoelectrode is similar to photocatalysis, step (1) and (2) are photon absorption and charge separation, which can result in oxidation (9–11) and reduction (3–6) reactions at the surface. The radicals produced will again depend on the potential at the band edges. When examining the degradation of dyes/photosensitive compounds, there is an additional mechanism involved (12), and the photoexcitation of the compound can increase the observed current due to the electron transfer from the excited compound to the photoanodes' conduction band and can give a false impression of visible light activity [32]; again, care must be taken when reporting the results when using dyes/photosensitive compounds. The electric field produced by external bias causes electrons to migrate towards the conductive support and holes to move towards the solution interface, thereby increasing the lifetime of the electron–hole pairs, which results in improved reaction rates [33]. At the counter electrode, additional ROS can be produced, helping to further enhance the PEC process. The electric field also results in the electrophoretic movement of charged species towards the oppositely charged electrode.

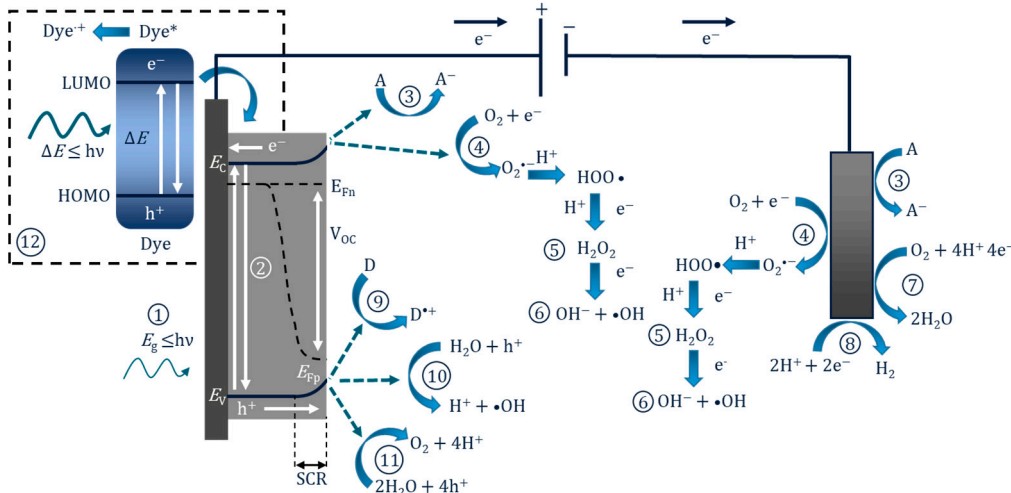

**Figure 3.** Diagram of the PEC process and pathways for radical production using a photoanode and a non-semiconducting counter electrode. (1) Photon absorption, (2) photoexcitation and recombination, (3) electron transfer to an electron acceptor, (4) oxygen reduction to superoxide, (5) formation of hydrogen peroxide, (6) formation of hydroxyl radical, (7) oxygen reduction to water, (8) proton reduction to hydrogen, (9) donor electron transfer, (10) oxidation of water to form hydroxyl radical, (11) oxygen evolution reaction, (12) dye sensitisation (* excited state) and electron transfer to the conduction band.

Figure 3 also shows the energy bands for an n-type semiconductor. Electrons (which are also Fermi particles) follow Fermi statistics, and in a semiconductor the statistical electrochemical potential of electrons is represented by the Fermi level ($E_f$). There is also a 50% probability of this energy level being occupied at any time. As the majority of carriers in n-type materials are electrons, the Fermi level is located close to the conduction band [29]. At the electrode–semiconductor interface under irradiation but with no bias (open circuit), the Fermi level (of an n-type material) is normally higher than the redox potential of the electrolyte, and this results in electrons being transferred to the solution to reach an equilibrium, and the positive charges generated at the surface of the photoelectrode create a space charge region (SCR), also known as the depletion layer, and this results in band bending [34]. Band bending and the formation of the SCR can also be induced by applying a positive bias to an n-type material. Band bending is important as it will aid the transfer and separation of the electron–hole pair. The SCR near the surface of the electrode has an increase in the minority carriers, and this increase results in the splitting of the Fermi level into two different electrochemical potentials for electrons ($E_{fn}$) and holes ($E_{fp}$), known as "*quasi-Fermi levels*"; the levels are positioned in the following order: $E_{fn} < E_f < E_{fp}$. The quasi-Fermi level results in a built-in electric field, which generates what is termed a "*photovoltage*" or "*photopotential*" or an "*open-circuit voltage*" ($V_{oc}$). The $V_{oc}$ value is the potential difference between the quasi-Fermi levels ($E_{fn}$-$E_{fp}$) [35]. Further information on electrodes and energy levels can be found in the book by Sato [29], as well as the reviews by Jiang et al. [35] and Zhang et al. [36].

## 4. Photoelectrodes

For photocatalytic applications, the band edge potentials should enable the generation of both ●OH and $O_2^{\bullet-}$; however, with PECs, the generation of both may not be as critical, as the counter electrode can be used to facilitate one side of the redox reaction. For example, a photoanode should have a band edge potential that enables the generation of ●OH, and the counter electrode can be used to enable the electron transfer to the solution (ideally forming a ROS). The reduction reaction is also assisted by the applied bias, completing the redox reaction. Therefore, this enables the use of smaller bandgap materials such as $WO_3$ to be used as a photoanode. A selection of different semiconducting materials with the bandgap/edges is shown in Figure 4.

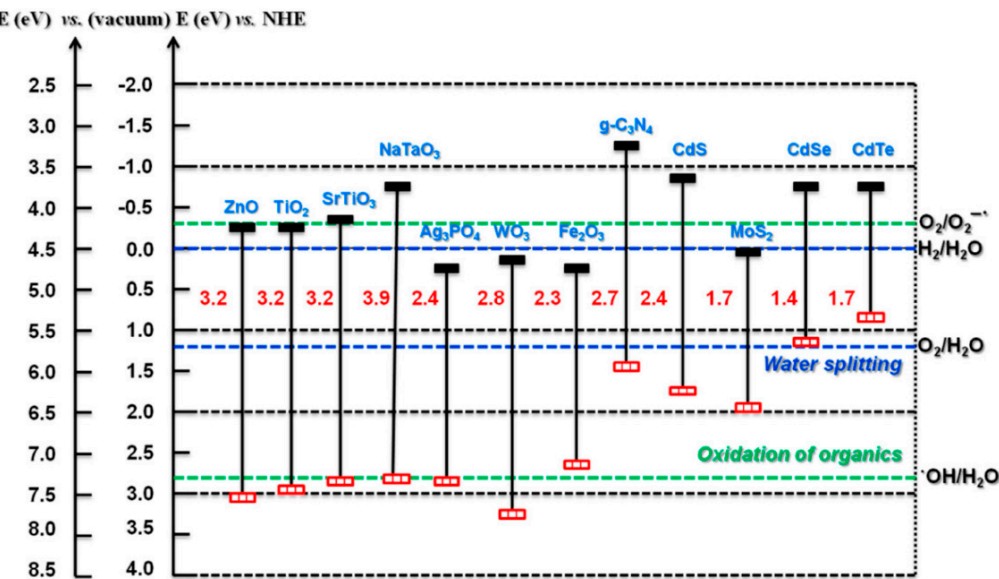

**Figure 4.** Bandgap and band edge potentials for different semiconductors. (Reproduced with permission from [37]: Kumar, S.; Karthikeyan, S.; Lee, A. g-C$_3$N$_4$-Based Nanomaterials for Visible Light-Driven Photocatalysis. *Catalysts* **2018**, *8*, 74, doi:10.3390/catal8020074. Copyright (2018), MDPI (Basel, Switzerland)).

Having suitable band edges is only one important factor. The semiconductor must also be photoactive and (electro)chemically stable in aqueous environments. It is also important to address the internal mechanism, to further explain the limiting factors of the material and therefore offer insights into how best to improve the semiconductor. The sequential internal mechanism (photon absorption, separation, charge transport/diffusion, catalytic efficiency and mass transport) has been compiled in a review by Takanabe [38] and is summarised in Figure 5. One aspect not included in the review is the growing area of crystal facet engineering, as the photocatalytic activity, surface absorption, surface energy and the valence band have been reported to vary between facets on a single crystal; thus, it is of interest to increase the area of the more effective facets. Further information on facet engineering and perspectives can found in the review by Tu et al. [39].

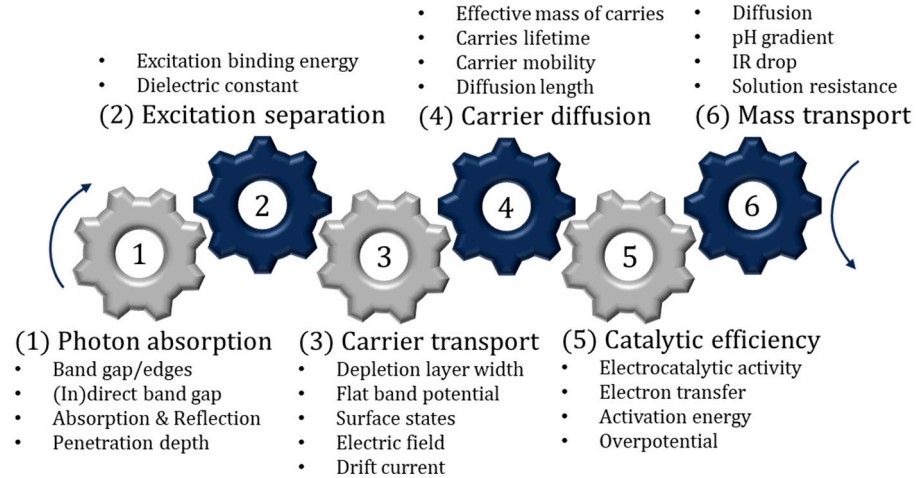

**Figure 5.** Sequential steps associated with photocatalysis and PECs.

Further enhancement has seen the combination of two materials being examined to enhance photoactivity. Depending on the band edge positions between the two materials, there can be three types of heterostructures, type I, II and III (Figure 6). The most promising is the z-scheme, which is a type II heterostructure using a p- and n-type material (p-

n junction) and, critically, for a solid-state system, an ohmic contact separates the two photosystems, essentially mimicking natural photosynthesis. The ohmic contact can be defined by a material that has a smaller work function ($\Phi$) than the semiconducting material ($\Phi_{ohmic} < \Phi_{SC}$); however, it has been reported that when using an additional semiconductor to prevent recombination, the work function of the ohmic contact should be between the two semiconductors, i.e., $\Phi_{SC1} < \Phi_{ohmic} < \Phi_{SC2}$ [40]. Therefore, the Fermi levels should also be in the same order with values that are close to one and another; this also known as Fermi level matching. This is important as, according to energy band theory, the electrons will flow from a higher Fermi level to a lower Fermi level and therefore the Fermi level matching should improve charge transport [41]. Examples of ohmic contacts that have been used included noble metals, Au [42], Ag [43], Pt [40], as well as reduced graphene oxide [44] and carbon nanotubes [45]. It is also important to investigate and account for the internal mechanism previously described, particularly at the material interfaces, as this will determine if recombination occurs and if the system is a type II heterostructure or true z-scheme.

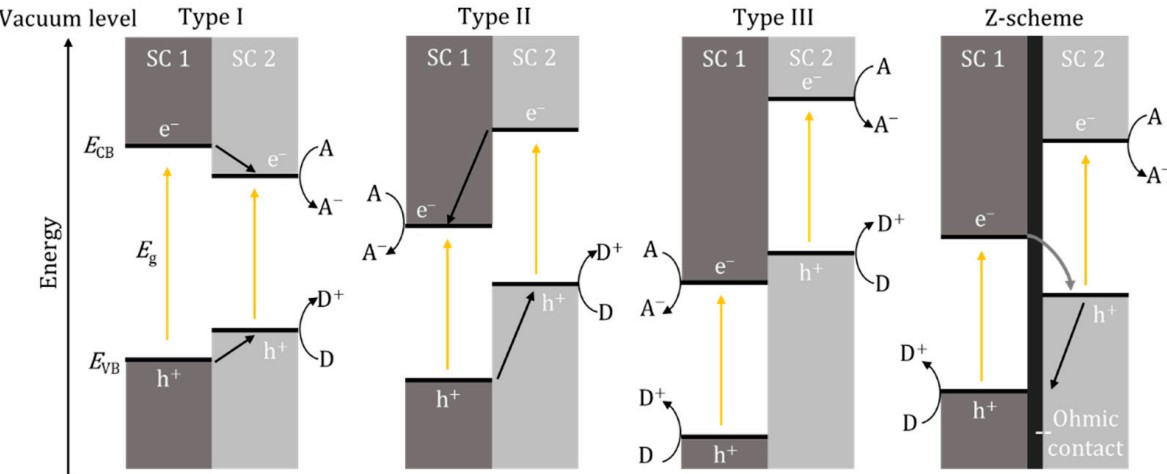

**Figure 6.** Band edge positions and electron/hole movement for type I, II and III heterostructures and z-scheme. SC—semiconductor, A—electron acceptor, D—electron donor, e⁻—electron, h⁺—hole.

## 5. Photoelectrocatalytic Reactor Design

Eleven distinct categories of PEC reactors are identified in this review; though different, there are some key defining parameters of any PEC reactor, i.e., the irradiation of the photoelectrode, which can be either back-face or front-face (Figure 7). For back-face irradiation, a conductive transparent supporting substrate is required (e.g., FTO), while with front-face irradiation, any suitable conducting supporting material can be used. Pablos et al. [46] demonstrated improved rates of *E. coli* inactivation when using a front-face configuration compared to a back-face one, using a TiO$_2$ (P25) particulate film on ITO with a Ni mesh counter electrode. The improvement was attributed to the mechanism of direct photoinactivation of *E. coli* by UV irradiation, ROS generated by PECs and the impact of hole mobility when using back-face irradiation. However, for methanol oxidation, back-face irradiation showed slightly improved kinetics, which was attributed to the porosity of the TiO$_2$ particulate film that enables the diffusion of methanol, a known hole scavenger, from the aqueous phase into the porous film and thus the limitations of hole mobility were less profound. Therefore, the optimal orientation of the photoanode may be dependent on the required application and characteristics of the water. For example, back-face irradiation should be used where the water has high turbidity or low UV transmission.

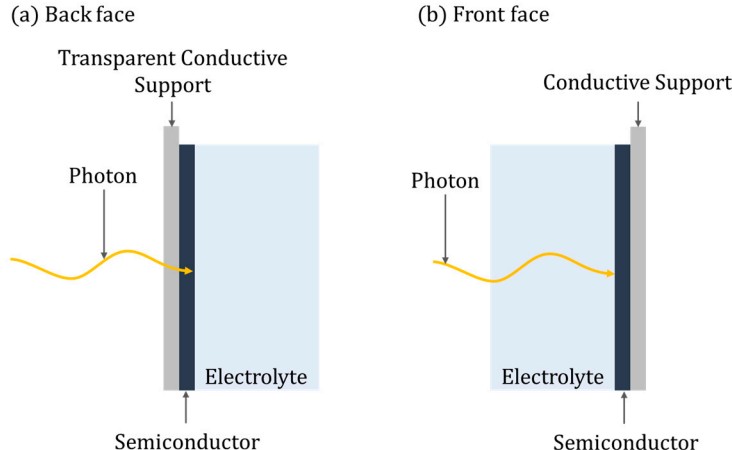

**Figure 7.** Photoelectrode ordination, (**a**) back face, (**b**) front face.

Another key parameter of any PEC reactor is the irradiation source, such as UV fluorescent lamps [46,47]; alternatively, more environmentally friendly and efficient UV LEDs have been used but are more expensive [48,49]. The advantage of using artificial irradiation is the ability to operate at any time with a uniform and consent photon flux. To avoid the associated costs, solar energy can be used as a free energy source and presents a truly clean technology. To date, there have been very few real solar experiments reported using PECs, although xenon lamps that can simulate the solar spectrum have been used [50]. The problem associated with real solar irradiation is the daily variation in solar irradiance, leading to variation in treatment times and a lack of efficient or suitable reactor designs. Moreover, for a PEC system to be a truly *"clean"* method of water treatment, the input energy (irradiation and external electrical bias) should be derived from a renewable source, e.g., solar or wind.

The remaining design parameters are geometrically and operationally based; most of the PEC reactors in this review used an n-type photoanode (commonly titania) and an inert counter electrode. The different reactor categories are discussed in the following segments. The operational parameters and selected results of the reactors are summarised in Appendix A, Table A1.

### 5.1. Experimental Reactors for Photoelectrode Testing

For the initial examination of photoelectrodes, small-scale experimental reactors are required to adequately control the reaction parameters, e.g., irradiation profile, temperature, mass transport, aeration, etc. A common experimental reactor is a two or three electrode system using a stirred tank reactor, commonly with a water jacket to regulate temperature [50–53]. A typical set-up is displayed in Figure 8a. It can utilise both front-face and back-face electrodes, enables air or gas sparging, mixing can be achieved with magnetic stirring and temperature control via the use of the water jacket and samples can be taken from the reactor.

Pablos et al. [54] used a reactor akin to the one in Figure 8a and reported that when comparing different photoanode materials, the magnitude of the photocurrent does not correlate to the rate of disinfection, i.e., titania nanotubes with a photocurrent of 175 µA gave a 5-log reduction of *E. coli* in 120 min but nitrogen-doped nanotubes with a lower photocurrent of 165 µA achieved 5-log reduction of *E. coli* in only 60 min. This can be attributed to the complex cycling of electrons between the defect states and the mid-gap states introduced by N-doping, and the resulting reductive reactions which can take place on the photoanode, and it was suggested that some of the electrons are transferred to molecular oxygen to form superoxide radical anions [55].

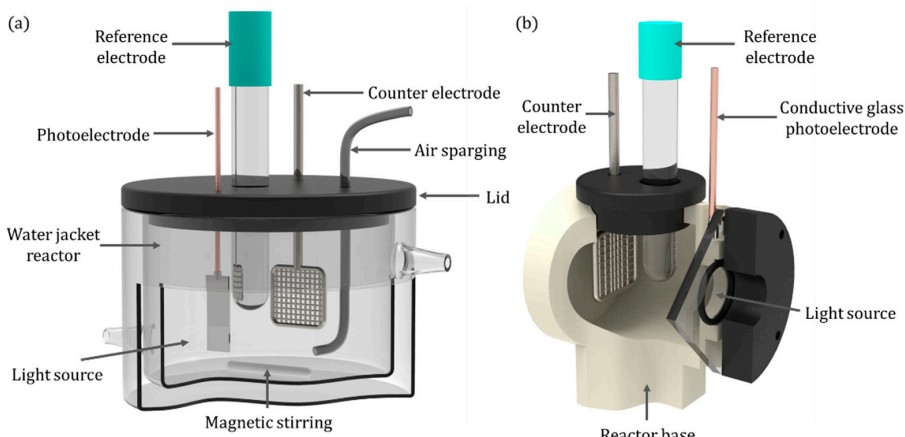

**Figure 8.** (**a**) Quartz water-jacketed PEC; (**b**) a back-face irradiated reactor.

For back-face irradiated electrodes, an alternative design can be used such as the stirred tank reactor used by McMurray et al. [56]. It enables the maximum irradiation to be absorbed by the photocatalyst film, and a stainless steel propeller was used for both mixing and as the counter electrode by using a carbon bush to make an electrical contact. Alternatively, a more traditional photoelectrochemical cell, as shown in Figure 8b, can be used for electrochemical analyses.

The experimental reactors described can also be used for photoelectrochemical measurements with a potentiostat; for example, linear sweep and cyclic voltammetry, amperometry, voltammetry and transient photocurrent response. Additionally, electrochemical impedance spectroscopy (EIS) can be used to examine charge transfer and to determine the flat band potential through Mott–Schottky analysis [57]. When combined with a monochromatic source, photoelectrochemical measurements can be used to determine the spectral photocurrent response and the incident photon conversion efficiency (IPCE) [58].

### 5.2. Annular Cylinder Reactor

The use of fluorescent UV lamps has resulted in designs in which the lamp is placed in the centre of the reactor, leading to annular cylindrical reactor designs and can be originated either horizontally [46,59] or vertically [47]. This configuration ensures that photons emitted in all directions are utilised by the rector. The design of Pablos et al. [46] (Figure 9) aimed to scale up a particulate film of $TiO_2$ (P25) on ITO to be used as the photoanode for the treatment of wastewaters.

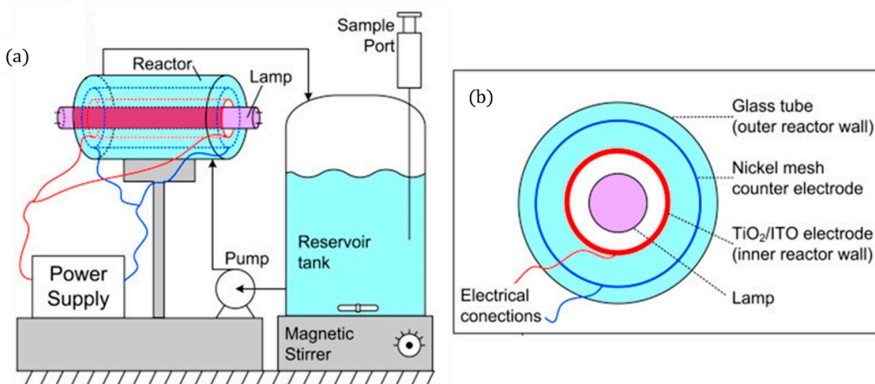

**Figure 9.** (**a**) Schematic representation of the experimental set-up, (**b**) PEC reactor cross-sectional view. (Adapted with permission from [46]: Pablos, C.; Marugán, J.; Adán, C.; Osuna, M.; van Grieken, R. Performance of $TiO_2$ photoanodes toward oxidation of methanol and *E. coli* inactivation in water in a scaled-up photoelectrocatalytic reactor. *Electrochim. Acta* **2017**, *258*, 599–606, doi:10.1016/j.electacta.2017.11.103. Copyright (2017), Elsevier (Amsterdam, The Netherlands)).

To ensure that the counter electrode (nickel mesh) is close to the photoanode without short-circuiting, 5 mm Teflon® spacers were used. Methanol oxidation shows no changes with different applied potentials but, for disinfection, the optimal potential was +1.4 V for a 3-log reduction of *E. coli* after 140 min to treat 1 L.

The inclusion of air/oxygen sparging directly into the reactor can take the form of an annular bubble column reactor, for example, Figure 10 is the design used by Kim et al. [47]. This has two purposes, firstly, the supply of oxygen for reductive pathways and, secondly, to introduce turbulence, enabling improved mass transport [60].

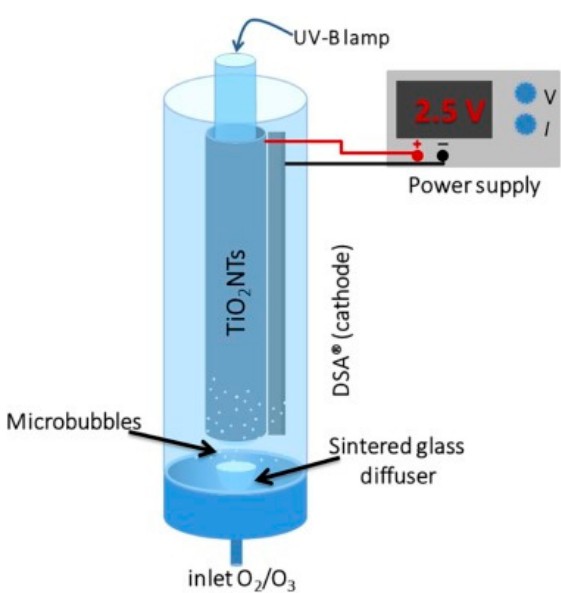

**Figure 10.** Schematic representation of the PEC annular bubble column reactor. (Reproduced with permission from [47]: Kim, J.Y.U.; Bessegato, G.G.; de Souza, B.C.; da Silva, J.J.; Zanoni, M.V.B. Efficient treatment of swimming pool water by photoelectrocatalytic ozonation: Inactivation of *Candida parapsilosis* and mineralization of Benzophenone-3 and urea. *Chem. Eng. J.* **2019**, 378, 122094; DOI:10.1016/j.cej.2019.122094. Copyright (2019), Elsevier (Amsterdam, The Netherlands)).

As titania nanotubes have been shown to improve photocurrent and degradation rates compared to a particulate film [54], it is therefore of interest to incorporate anodised $TiO_2$ nanotubes into a reactor, as with the reactor of Kim et al. Their system used a titanium cylinder with titania nanotubes, a DSA® (De Nora) cathode and a UVB lamp (36 W) to examine the degradation of benzophenone-3 (a sunscreen compound), urea and the fungal species *Candida parapsilosis* with the aim of treating swimming pool water. The system was purged with $O_2$ and/or with $O_3$, as ozone alone can be used for water treatment [61], its incorporation with PECs showed further improved degradation rates. With increasing applied bias of +1.0, +1.5 and +2.0 V, the degradation only improved slightly each time. After a 20 min treatment, the order of effectiveness for TOC reduction was PEC + $O_3$ > PEC > $O_3$. The standard figure of merit was used to examine the economics as a function of electric usage, as proposed by Bolton et al. [62] for first-order kinetic reactions with the units kW h m$^{-3}$ order$^{-1}$. The PEC results (58.2 kW h m$^{-3}$ order$^{-1}$) show the highest cost compared to PEC + $O_3$ (21.4) and $O_3$ (9.8), though the increased rate of PEC + $O_3$ is more than the increased cost compared to $O_3$. Further cost reduction can be achieved by using higher efficiency lighting, e.g.., LEDs, or the use of solar irradiation.

The cylindrical design of Montenegro-Ayo et al. [63] used titanium discs which were anodised to produce a titania nanotube array, the counter electrode was unmodified titanium discs and the irradiation source was a 14 W UVC lamp (275 nm). A schematic of the reactor is presented in Figure 11. The reactor was tested for the degradation of acetaminophen (10 mg L$^{-1}$) in 0.02 M $Na_2SO_4$. Different applied cell biases were examined

(0, 4, 8, 16, 32, 64 V) and, at +8.0 V, there was a 95% reduction after 5 h when operated under irradiation compared to 3% for only electrolytic and 72% for photocatalytic degradation (open circuit). Increasing the applied bias >+8.0 V did not offer a significant improvement in the rate but did significantly increased the energy requirement.

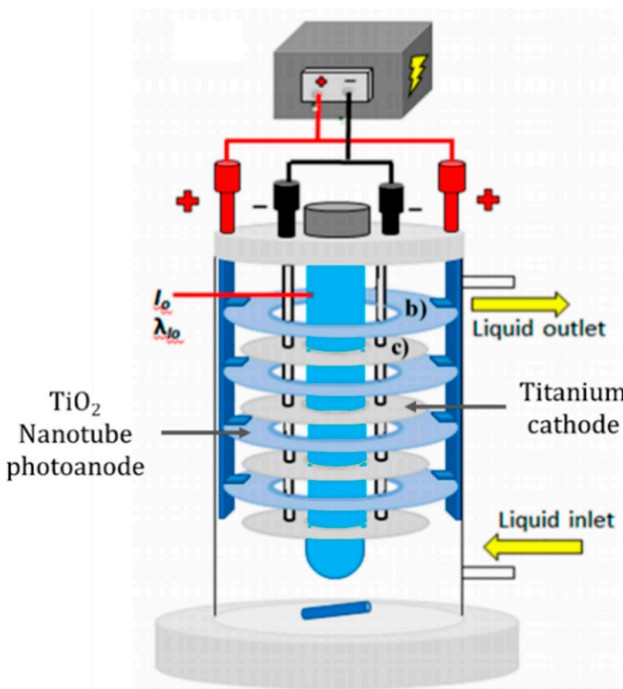

**Figure 11.** Schematic representation of the experimental set-up using anodised titanium photoanode. (Adapted with permission from [63]: Montenegro-Ayo, R.; Morales-Gomero, J.C.; Alarcon, H.; Cotillas, S.; Westerhoff, P.; Garcia-Segura, S. Scaling up Photoelectrocatalytic Reactors: A TiO$_2$ Nanotube-Coated Disc Compound Reactor Effectively Degrades Acetaminophen. *Water* **2019**, *11*, 2522, doi:10.3390/w11122522. Copyright (2019), MDPI (Basel, Switzerland)).

*5.3. Compound Parabolic Collector*

For real solar applications, cylindrical reactors can be combined with a solar concentrator to increase the irradiance directed into the reactor. Specifically, the compound parabolic collector (CPC) has the advantage of irradiating the tube with both direct and diffused irradiation regardless of the time of day without the need for solar tracking, enabling the full circumference of the reactor to be irradiated with a one-sun equivalent [64]. The two-compartment cell design of Fernandez-Ibañez et al. [65] used a CPC reactor, as displayed in Figure 12.

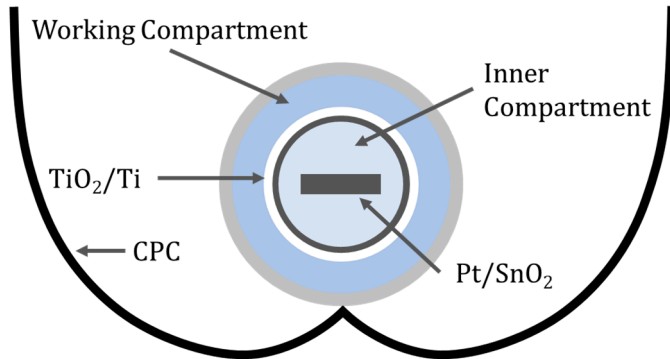

**Figure 12.** Schematic of the two-compartment reactor with a CPC.

The inner compartment contained 0.2 M $H_2SO_4$ (pH = 0.4) and the high concertation of protons can be reduced to form hydrogen at the cathode; in this case, a glass substrate coated with $SnO_2$ and Pt using a solution of $H_2PtCl_6$. The working compartment used 0.02 M $H_2SO_4$ (pH = 1.4) to degrade 4-chlorophenol and pyrimethanil with a dip-coated particulate film of $TiO_2$ onto titanium in which the film thickness was 100 nm and the average particle size was 27 nm. The photooxidation of 4-chlorophenol at +1.4 V (RHE) resulted in a normalised reduction rate of 7.3 mg $min^{-1}$ $m^{-2}$, this is 69 times the rate for a $TiO_2$ slurry reactor (0.106 mg $min^{-1}$ $m^{-2}$), and increasing to +1.8 V (RHE) resulted in a smaller reduction of 3.8 mg $min^{-1}$ $m^{-2}$, concurring with others that there is optimal bias when using PECs [66]. For pyrimethanil degradation at +1.5 V (RHE), the rate was 6.21 mg $min^{-1}$ $m^{-2}$, 59 times higher than the value found for $TiO_2$ slurries.

The design of McMichael et al. [67] (Figure 13) used a single-compartment reactor with a titania nanotube array produced by anodisation of a titanium mesh to be used as the photoanode, a carbon felt counter electrode placed in the centre of the reactor and an applied cell potential of +1.0 V.

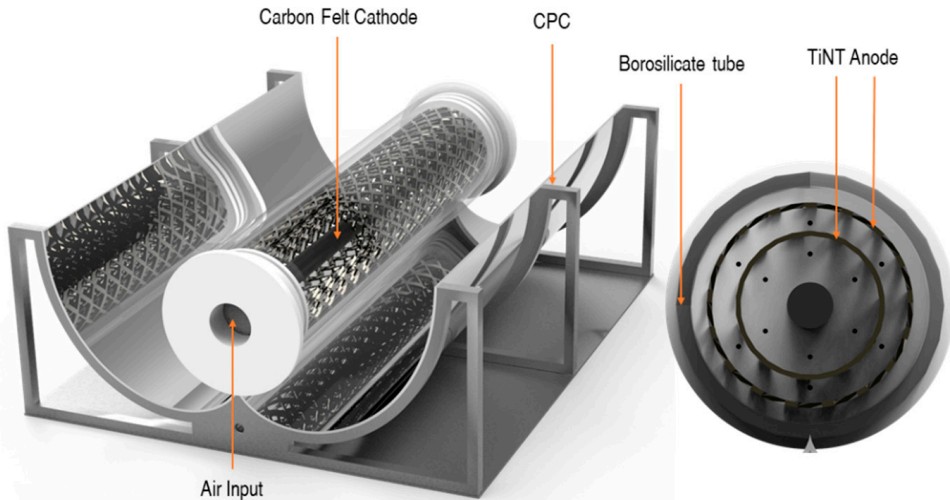

**Figure 13.** PEC reactor configuration with CPC. (Reproduced with permission from [67]: McMichael, S.; Waso, M.; Reyneke, B.; Khan, W.; Byrne, J.A.; Fernandez-Ibanez, P. Electrochemically assisted photocatalysis for the disinfection of rainwater under solar irradiation. *Appl. Catal. B* **2021**, *281*, 119485, doi:10.1016/j.apcatb.2020.119485. Copyright (2021), Elsevier Elsevier (Amsterdam, The Netherlands)).

The reactor was angled to maximise the solar irradiance at solar noon and purged with air from the bottom of the reactor. The reactor was tested under real solar irradiance with an average UV intensity of 1.12 mW $cm^{-2}$, for the inactivation of environmental strains of *E. coli* and *P. aeruginosa* using rainwater as the electrolyte which was autoclaved before use and had a conductivity of 70 $\mu S$ $cm^{-2}$. No additional electrolytes were added to the solution. The PEC reactor achieved a 5.5-log reduction for *E. coli* and a 5.8-log reduction of *P. aeruginosa*. The PEC reactor improved the performance of solar inactivation compared to a solar-only reactor of the same geometry that was tested in tandem. The inactivation of the microorganisms was also examined by molecular analyses using ethidium monoazide bromide, a nucleic acid-binding dye, as a pre-treatment before running a quantitative real-time polymerase chain reaction to examine if the membrane of the cell had been damaged. The PEC reactor showed a higher reduction in gene copies, demonstrating that the PEC process is more effective for the inactivation of microorganisms compared to solar only.

### 5.4. Flatplate or Sandwich Reactor

These reactors are categorised as having flat electrodes which are commonly parallel to each other, like the reactor design of Mousset et al. [68] shown in Figure 14. Their reactors used FTO glass coated with $TiO_2$ nanoparticles and a porous carbon felt counter electrode with a 6 W UVA (365 nm peak) lamp. The base of the reactor was 3D printed and used recirculating water, and the first reactor design sparged $O_2$ into a separate tank and the second and more effective reactor sparged $O_2$ directly into the reactor, and the improvement was attributed to the direct absorption of $O_2$ onto the carbon cathode, which can then be reduced, demonstrated by the increase in $H_2O_2$ generation. Other studies have shown improvement with increased airflow rate [69]. The results showed a $73.2 \pm 1.2\%$ reduction of phenol after 8 h when applying a fixed current density of 1.25 mA cm$^{-2}$ (PEC) compared to 44% for photocatalysis alone. When using a fixed current (galvanostatic), the potential must not rise significantly to the point at which damage can be caused to the photoelectrode [70]. The system was also examined using Fe(III)-hydroxy for the Fenton process and although it increased the rate of phenol degradation, for practical applications, it will require the removal of the iron catalyst, which defects the point using immobilised photocatalysts.

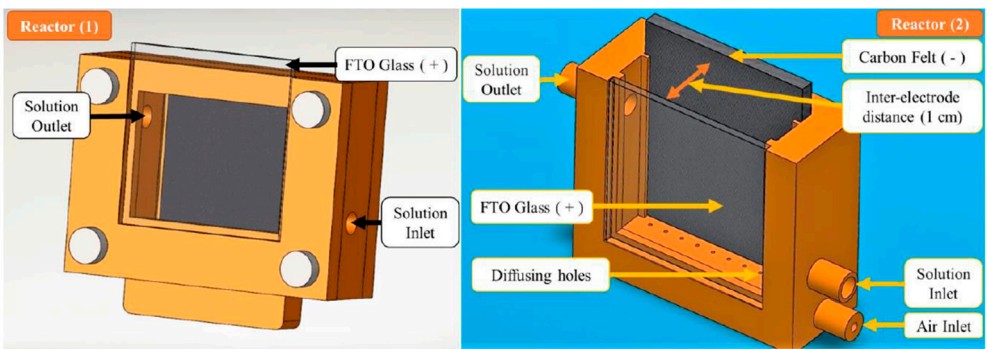

**Figure 14. Reactor (1)** air bubbled in a separate tank, **Reactor (2)** air bubbled directly into reactor. (Reproduced with permission from [68]: Mousset, E.; Huang Weiqi, V.; Foong Yang Kai, B.; Koh, J.S.; Tng, J.W.; Wang, Z.; Lefebvre, O. A new 3D-printed photoelectrocatalytic reactor combining the benefits of a transparent electrode and the Fenton reaction for advanced wastewater treatment. *J. Mater. Chem. A* **2017**, *5*, 24951–24964, doi:10.1039/c7ta08182k. Copyright (2017), Royal Society of Chemistry Publishing (Cambridge, UK)).

Bai et al. [71] examined the use of a thin cell reactor (Figure 15a), in which the thickness of the solution (distance from the surface of the electrode to the window) was 0.2 mm, minimising the UV absorption by the solution and maximising the surface to volume ratio. Compared to a conventional PEC reactor (Figure 15c), the thin cell reactor achieved a 54.8% reduction compared to 14.6% with an initial concentration of 120 mg L$^{-1}$ of tetracycline after 1 h, with a +2.0 V bias and a flow rate of 10 mL min$^{-1}$. The design also enabled irradiation from both sides of the photoelectrode (Figure 15b), which doubled the performance. Overall, the thin cell reactor confirms the need for a high surface to volume ratio, sufficient mass transfer, efficient use of photons and the need for an efficient photoanode.

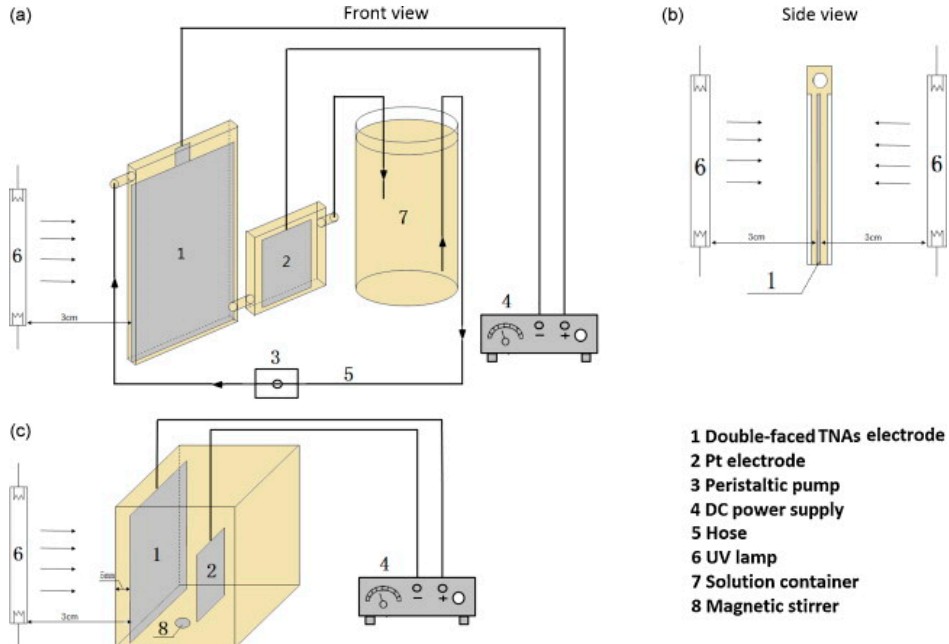

**Figure 15.** Schematic diagram of the TNA-based thin-layer PEC reactor (**a**,**b**) and the conventional PEC reactor (**c**). (Reproduced with permission from [71]: Bai, J.; Liu, Y.; Li, J.; Zhou, B.; Zheng, Q.; Cai, W. A novel thin-layer photoelectrocatalytic (PEC) reactor with double-faced titania nanotube arrays electrode for effective degradation of tetracycline. *Appl. Catal. B* **2010**, *98*, 154–160, doi:10.1016/j.apcatb.2010.05.024. Copyright (2010), Elsevier (Amsterdam, The Netherlands)).

The flatplate reactor of Byrne et al. [72] used a $TiO_2$ photoanode prepared by electrophoretic deposition of P25 on ITO-coated soda lime glass ($1.0 \pm 0.2$ mg cm$^{-2}$) and a Pt-coated Ti mesh counter, positioned parallel to each other, with a re-circulating solution at a rate of 60 mL min$^{-1}$ and two 18 W UVA lamps (peak 370 nm) as the irradiation source. The reactor was tested for the degradation of formic acid (3.18 mM) with $KNO_3$ as the supporting electrolyte (0.5 M pH = 3) and an applied potential of +1.0 V (SCE). The incident photon to current efficiency was 2.2% and the apparent quantum yield for the degradation of formic acid was 7%. Interestingly, the $O_2$ sparged solution resulted in lower photocurrent and reduction values compared to air sparged solutions.

### 5.5. Channel/Microchannel Reactor

The previously stated thin cell reactor had a high surface to volume ratio, as does the channel/microchannel reactor; however, the following category has a more distinct and defined style compared to the previous flat thin cell reactor. To increase the residence time and improve the mass transfer in the reactor, channels can be incorporated into the design (Figure 16).

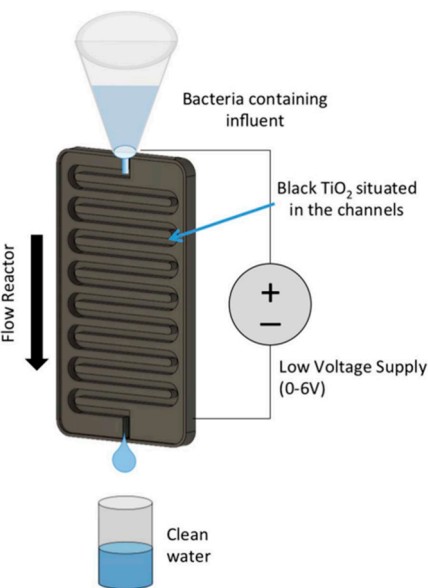

**Figure 16.** Schematic of a channel reactor using titanium nanotubes annealed in 2% hydrogen. (Reproduced with permission from [70]: Carlson, K.; Elliott, C.; Walker, S.; Misra, M.; Mohanty, S. An effective, point-of-usewater disinfection device using immobilized black $TiO_2$ nanotubes as an electrocatalyst. *J. Electrochem. Soc.* **2016**, *163*, H395–H401, doi:10.1149/2.0651606jes. Copyright (2016), IOP Publishing (Bristol, UK)).

The design of Carlson et al. [70] (Figure 16) used a titania nanotube photoelectrode annealed in 2% hydrogen in a channel reactor and a flow rate of 50 mL min$^{-1}$ which leads to a contact time of 25 s. To increase the rate of disinfection, 50 mg L$^{-1}$ NaCl was added to natural lake water which increased the conductivity and promoted the production of chorine radicals which was beneficial to the disinfection process [73]. The system was operated at +6.0 V as increasing to +7.0 V caused the nanotubes to delaminate from the surface, and no bacteria were detected with and without 100 mWcm$^{-2}$ of solar irradiation with an initial *E. coli* concentration of 165 CFU 100 mL$^{-1}$. The potential used exceeds the forward voltage of $TiO_2$, hence the irradiation only increasing the current density by 34 μAcm$^{-2}$/7.7% compared to the dark current (404 μAcm$^{-2}$); thus, the authors dubbed the reactor as an electrocatalytic reactor rather than a PEC reactor.

Alternatively, a microchannel reactor (Figure 17) can be used, in which the channel is 1–1000 μm; this has the advantages of having a significantly larger surface to volume ratio (10–30 cm$^2$cm$^{-3}$) compared to other PEC reactors (0.7–5 cm$^2$cm$^{-3}$ [46]) and fine flow control using inlet and outlet branches to ensure a uniform flow and, therefore, achieving a near consent residence time [74], there is also improved mass transport and direct photon absorption at the surface of the photoelectrode with minimal photon absorption by the solution [49]. However, these reactors can be difficult to scale up, resulting in non-uniform residence times and, for real water sources, particulates of a set size would have to be filtered to prevent blockages within the microchannels [75]. The design of Wang et al. [49] used a thin film of monoclinic $BiVO_4$ (n-type) on ITO as the photoelectrode with an ITO counter electrode for the degradation of methylene blue, and 0.1 M NaCl as the electrolyte and a flow rate of 75 μL min$^{-1}$ were used, and the seemingly small flow rate is due to reactor size (1 cm × 1 cm). As an example, scaling up to 20 cm × 20 cm would results in a flow rate of 1.8 L h$^{-1}$. The flow rate of any flat microchannel reactor can be easily calculated using Equation (6), where Q is the flow rate, S is the surface area, H is the height of the reactor and t is the residence time.

$$Q = SH/t \tag{6}$$

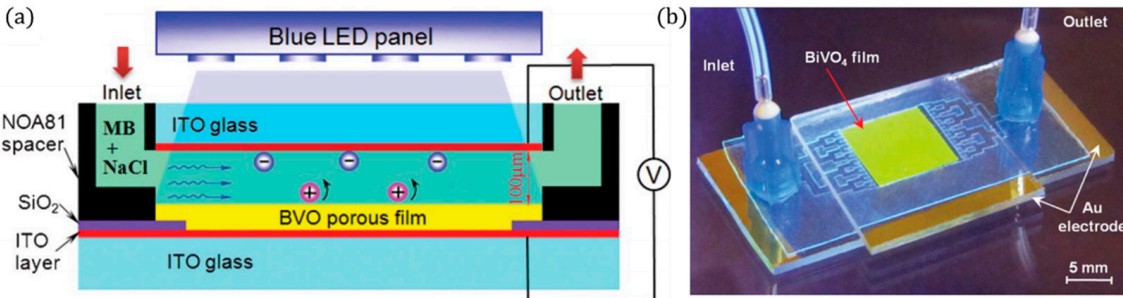

**Figure 17.** (**a**) Cross-sectional view of the reaction chamber, (**b**) photo of the PEC microreactor. (Adapted with permission from [49]: Wang, N.; Zhang, X.; Chen, B.; Song, W.; Chan, N.Y.; Chan, H.L.W. Microfluidic photoelectrocatalytic reactors for water purification with an integrated visible-light source. *Lab Chip* **2012**, *12*, 3983–3990, doi:10.1039/c2lc40428a. Copyright (2012), Royal Society of Chemistry Publishing (Cambridge, UK)).

A blue LED (peak 402 nm) was used as a compact light source, with high energy efficiency, low heating and which delivers direct and uniform irradiation, and the intensity can be controlled by the applied voltage to the LED, so at 10 V, the power density was 80 mWcm$^{-2}$, and increasing to 11 V resulted in 140 mWcm$^{-2}$ but also results in heating (>30 °C). The microchannel reactor was tested with a range of different cell voltages ($-2.2$ V to $+2.2$ V), and the optimal anodic potential was $+1.5$ V, with higher potentials resulting in lower degradation rates. The negative biases exhibited higher performance, i.e., when used as a photocathode (N.B. no other reports on BiVO$_4$ as a (photo)cathode could be found), and this was attributed to the higher electrical current. O$_2$ reduction to form ROS is the dominating pathway for degradation, and chloride ions scavenge the ●OH, leading to weaker chloride radicals, and using the BiVO$_4$ to reduce oxygen is more effective than ITO.

Microchannel reactors have also been used for the inactivation of *E. coli* [48,76]. The design of Liu et al. [48], with a channel depth of 250 μm, examined both a particulate and titania nanotubes as the photoanode, with a Pt mesh auxiliary electrode and a UV LED array (8 mWcm$^{-2}$, 365 nm) and using a 0.1 M NaNO$_3$ electrolyte. The potential was fixed at $+0.7$ V (Ag/AgCl) as this was the point of photocurrent saturation for the nanotube photoelectrode. The nanotube array reached the detection limit with a resistance time of 97 s compared to 311 s with the particulate film (based on the method reported, this would be a ~6-log reduction. N.B. a reduction in microorganisms should ideally be reported in log$_{10}$ scale). The nanotube electrode was reused 10 times with no distinguishable effect on performance, therefore demonstrating the stability of the electrode.

*5.6. Rotating Electrode Reactor*

The use of a rotating electrode in a reactor improves the mass transport by introducing turbulence/mixing and more contact between the contaminant and electrode(s), which overall improves the performance of the PEC reactor [77]. The self-rotating design of Cho et al. [78] moves both electrodes on a central axis in a tubular reactor, as shown in Figure 18. Titania nanotubes as the photoanode and a titaniumcounter for the removal of methylene blue were used. The results from the experiments showed the importance of reducing the IR drop, by minimising the distance between electrodes, which subsequently improves the rate of degradation, as well as the importance of mass transport and the retention time, as the rotation speed was dependent on the inlet flow rate, the optimum thereof occurring at 90 RPM, but with a higher RPM, the retention time is lower due to the increased flow rate which results in lower rates of degradation.

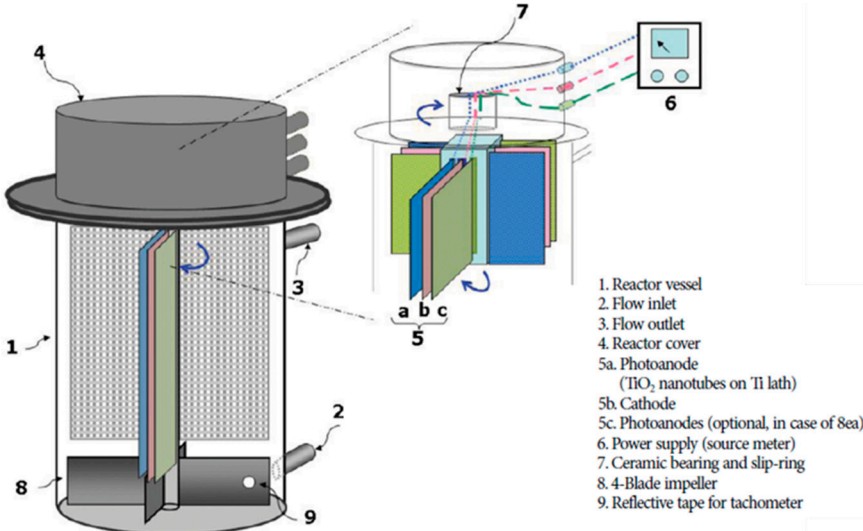

**Figure 18.** Rotational reactor using titanium nanotube photoanode and titanium cathode. (Adapted with permission from [78]: Cho, H.; Seo, H.S.; Joo, H.; Kim, J; Yoon, J. Performance evaluation of rotating photoelectrocatalytic reactor for enhanced degradation of methylene blue. *Korean J. Chem. Eng.* **2017**, *34*, 2780–2786, doi:10.1007/s11814-017-0198-7. Copyright (2017), Springer Nature (Seoul, Korea)).

Xu et al. [26] used a rotating disk reactor (RDR) with a $TiO_2$ photoelectrode prepared via a sol–gel method, with a copper cathode, in a solution of 3.5 mM $Na_2SO_4$ for the degradation of Rhodamine B, with a UV lamp (256 nm, 15 mWcm$^{-2}$), as depicted in Figure 19. The photoelectrode is only partially submerged in the solution, the bottom segment acts as a conventional PEC cell and the top unsubmerged segment forms a thin film of the solution/contaminants on the surface of the photoelectrode and is exposed to air and the full irradiation. Increasing the rotation (RPM) improved the rate of Rhodamine B degradation up to 90 RPM, while other RDRs have shown improvement with increased RPM [79,80]. The RDRs also show improvement compared to a conventional PEC cell, and the improvement is also more profound with higher contaminant loadings [26,79,80]. Further improvement has been achieved by using an electrical discharge linear cutting machine to create a 3D pyramid structure (2, 4 and 6 mm height) on a titanium substrate to increase the geometric surface area, followed by dip coating to coat the substrate with $TiO_2$ [79]. A titanium nanotube RDR has also been used due to the enhancement that the 1D material has shown compared to particulate films [81].

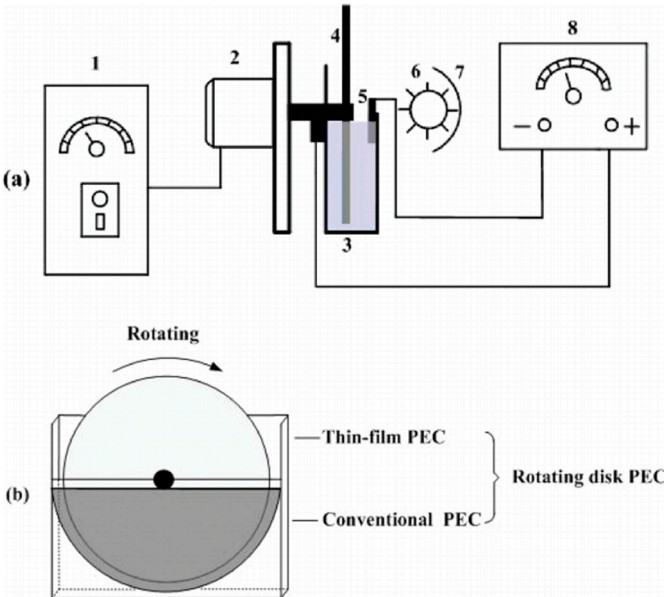

**Figure 19.** (**a**) Schematic diagram of the side view of the $TiO_2$/Ti rotating disk PEC reactor. The figure is not to scale. 1, speed controller; 2, motor; 3, electrolytic cell; 4, $TiO_2$/Ti rotating disk anode; 5, cathode; 6, UV lamp; 7, aluminium foil; 8, DC power supply. (**b**) The front view of the $TiO_2$/Ti rotating disk electrode. (Reproduced with permission from [26]: Xu, Y.; He, Y.; Cao, X.; Zhong, D.; Jia, J. $TiO_2$/Ti rotating disk photoelectrocatalytic (PEC) reactor: A combination of highly effective thin-film PEC and conventional PEC processes on a single electrode. *Environ. Sci. Technol.* **2008**, *42*, 2612–2617, doi:10.1021/es702921h. Copyright (2008), American Chemical Society (Washington, DC, USA)).

### 5.7. Optical Fibre Reactor

Several articles have reported using optical fibres in photocatalytic reactors [82–84], however, for PECs, only one such reactor has been reported [85]. The use of optical fibres has the advantage of transporting light through the length of the fibre, improving the light illumination efficiency and increasing the surface area by using bundles of coated fibres [84]. As optical fibres are used to transfer phonons considerable distances (in the Km range), the fibre needs to be treated (mechanically or chemically) to induce holes/cracks on the surface for the light to escape and be absorbed by the photocatalyst [82]. To use the fibre as a photoelectrode, a conductive film needs to be applied. The design of Esquivel et al. [85] (Figure 20) used $SnO_2$:Sb as a transparent conductive film with electrophoretically deposited $TiO_2$, and carbon cloth threads were used to increase the conductivity during the deposition as the $SnO_2$:Sb had a high resistance of 0.1 MΩ; alternatively, ITO has also applied to optic fibres [86]. The reactor used a carbon cloth cathode, with 0.05 M $Na_2SO_4$ adjusted to pH 3, a fixed current of 1.0 mAcm$^{-2}$ was applied after 3 h and 80 mg L$^{-1}$ of $H_2O_2$ was produced, after which 15 mg L$^{-1}$ of Azo Orange II dye was added. After 60 min, a 50% decolourisation was observed when using PECs, though electrolytics showed comparable results. When analysing the total organic carbon (mineralisation), the PECs showed distinguishable results ($k$ = 0.0055 min$^{-1}$) compared to electrolysis ($k$ = 0.0035 min$^{-1}$), demonstrating that decolourisation is not the best method for evaluating reactors, as reported in other articles [22,87].

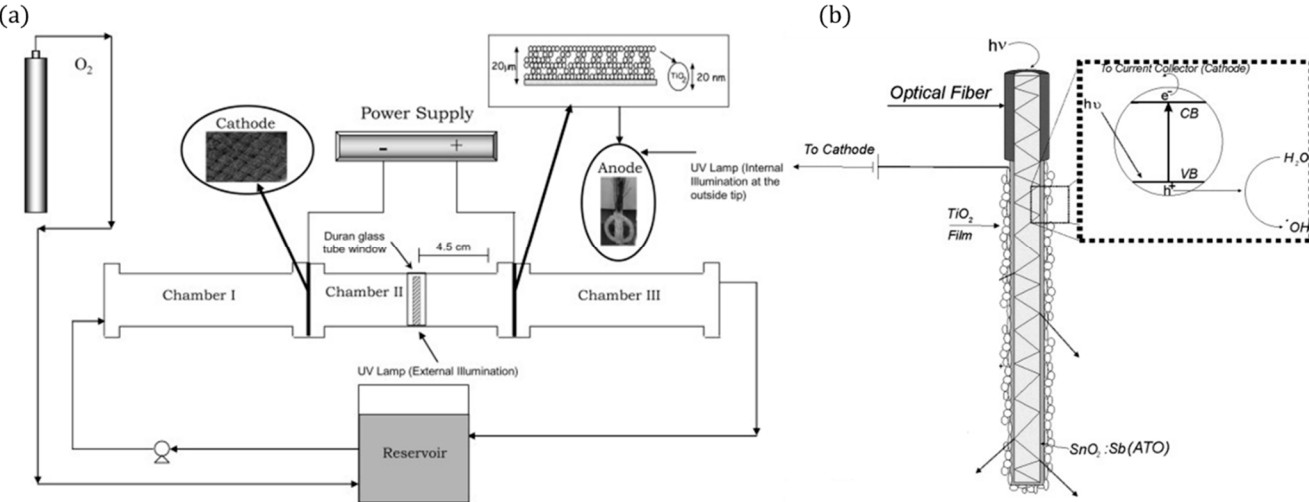

**Figure 20.** (**a**) Schematic representation of the experimental set-up, (**b**) fibre optic photoanode. (Adapted with permission from [85]: Esquivel, K.; Arriaga, L.G.; Rodríguez, F.J.; Martínez, L.; Godínez, L.A. Development of a TiO$_2$ modified optical fiber electrode and its incorporation into a photoelectrochemical reactor for wastewater treatment. *Water Res.* **2009**, *43*, 3593–3603, doi:10.1016/j.watres.2009.05.035. Copyright (2009), Elsevier (Amsterdam, The Netherlands)).

### 5.8. Membrane Filter Reactor

Photocatalytic materials have been added to membranes to reduce the effect of membrane fouling, increasing the lifetime of the filter and by extension improving the effective cost per volume, as well as enabling the removal of contaminants that are smaller than the pore size of the membrane [88]. These systems require the use of a conductive membrane or a secondary material to increase the conductivity, for example, carbon nanotubes [89], graphitic carbon [90], PEDOT [91] or stainless steel mesh sandwiched in a nylon fibre membrane [92] have been examined to facilitate electron transport, enabling the membrane to be used as a photoelectrode.

The design of Wang et al. [89], shown in Figure 21, used a porous ceramic membrane made from Al$_2$O$_3$.

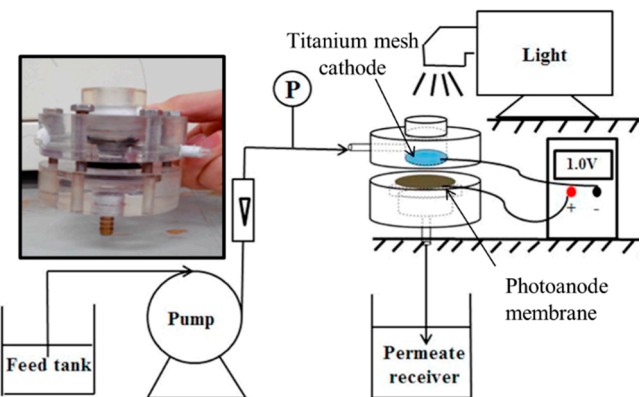

**Figure 21.** Experimental set-up, insert photo of the reactor. (Adapted with permission from [89]: Wang, X.; Wang, G.; Chen, S.; Fan, X.; Quan, X.; Yu, H. Integration of membrane filtration and photoelectrocatalysis on g-C$_3$N$_4$/CNTs/Al$_2$O$_3$ membrane with visible-light response for enhanced water treatment. *J. Membr. Sci.* **2017**, *541*, 153–161, doi:10.1016/j.memsci.2017.06.046. Copyright (2017), Elsevier (Amsterdam, The Netherlands)).

To increase the electrical conductivity, carbon nanotubes were added to the membrane with graphitic carbon nitride used as the semiconducting material for the degradation of phenol (5 mg L$^{-1}$). After loading the additional materials on the membrane, the mean pore

size was 297 nm; for comparison, phenol has an effective molecular diameter of 0.75 nm [93]. The cathode was a titanium network, to which irradiation of 100 mWcm$^{-2}$ (>400 nm) was directed and passed through to the membrane. The degradation experiments showed a 7% reduction in phenol with the membrane only, and the reduction was attributed to surface absorption, and when applying irradiation, the reduction increased to 26%, and applied bias of +0.5 V, +1.0 V and +1.5 V resulted in a reduction of 29%, 71% and 94%, respectively; the improvement is attributed to the generation of •OH and reduced recombination due to the applied bias. Humic acid was used to examine the anti-fouling capability of the membrane, and by examining the permeability (L m$^{-2}$ h$^{-1}$ bar$^{-1}$), the highest permeability was observed when operated at +1.5 V under irradiation, ~3 times higher than the membrane only, demonstrating the anti-fouling capabilities of the PEC membrane, as demonstrated by others [90,91].

The PEC membrane reactors can also be submerged in the solution, like the design of Gao et al. [94], which can be operated in either a recirculating mode (Figure 22) or a continuous influence and effluent filter mode. In this design, the photoelectrode is a composite, where carbon fibre felt acts as the framework, with one side coated with ZnIn$_2$S$_4$ microspheres (0.4–0.8 µm) and the reverse side is coated with polyvinylidene fluoride, which increases the tensile strength of the composite and ensures that the solution passes thought the photoactive layer. The reactor was tested for the removal of tetracycline (4 mg L$^{-1}$) with a titanium plate counter electrode. The optimal bias was −0.3 V, and the cathodic bias could generate O$_2^{•-}$, which indicates the O$_2^{•-}$ is more beneficial for the degradation of tetracycline, which has also been observed for methylene blue with WO$_3$ [49]. The circulating operation had a removal rate of 87% after 3 h, and under constant operation (24 ± 0.1 L m$^{-2}$ h$^{-1}$), the average reduction was 71% in the first 12 h run, so additional runs were conducted to assess the stability; however, the subsequent runs had lower reduction rates compared to the initial run. This was attributed to absorption on the surface during the first run, resulting in a higher reduction; once the maximum absorption is achieved, the reduction is primarily via degradation.

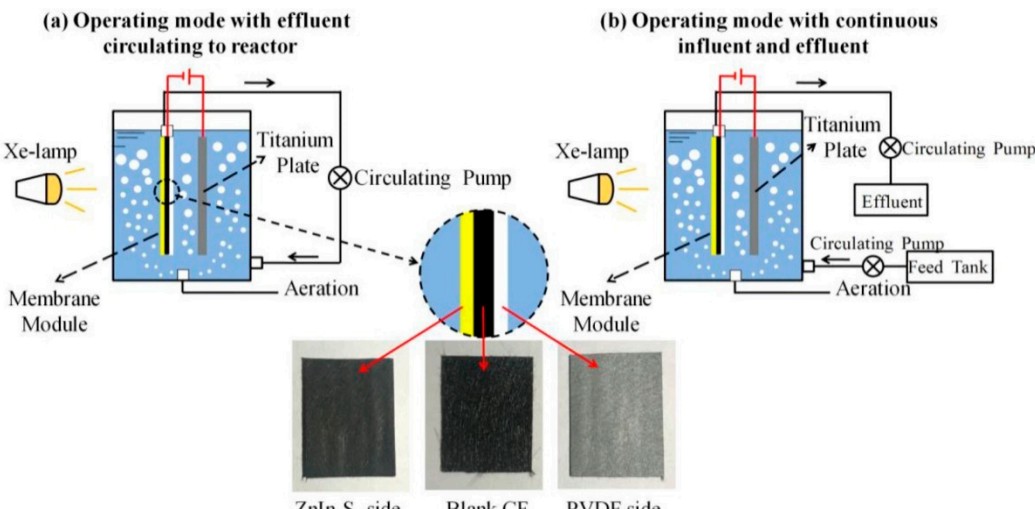

**Figure 22.** Schematic of the submerged photoanode membrane and separate counter electrode. (Adapted with permission from [94]: Martins, A.S.; Lachgar, A.; Boldrin Zanoni, M.V. Sandwich Nylon/stainless-steel/WO$_3$ membrane for the photoelectrocatalytic removal of Reactive Red 120 dye applied in a flow reactor. *Sep. Purif. Technol.* **2020**, *237*, 116338, doi:10.1016/j.seppur.2019.116338. Copyright (2020), Elsevier (Amsterdam, The Netherlands)).

### 5.9. Two-Compartment Reactor

The electrodes can be separated into individual compartments, as shown in Figure 23. The separation can be done by using a salt bridge, an ion exchange membrane or, as with the previously discussed cylindrical CPC design of Fernandez-Ibañez et al. [65], a glass frit

filled with an ionic conductive polymer (Nafion solution) to form the two-compartment reactor; however, these reactors are less commonly reported than the one-compartment reactor for degradation and disinfection. The two-compartment cell is more commonly reported for gaseous reactions to separate reactants and products, i.e., $O_2$, $CO_2$, $CH_4$, $H_2$ [95].

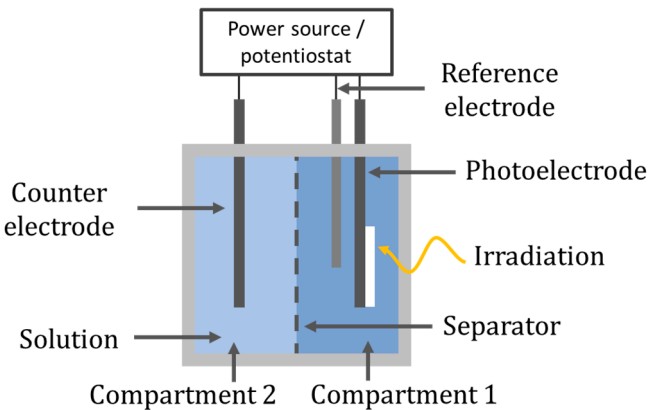

**Figure 23.** Two-compartment reactor set-up.

The work of Leng et al. [96] demonstrated that using a one-compartment had a higher rate of aniline degradation compared to a two-compartment reactor which was separated using a low porosity glass frit, a $TiO_2$/Ti photoanode prepared via dip coating and a silver sheet counter electrode. The working volume was the same in both set-ups; however, in the two-compartment reactor, only the photoanode was used for aniline degradation, while in the single set-up, both electrodes can be used to generate ROS for the single purpose of aniline degradation; therefore, it would be surprising if the one-compartment reactor was not more effective. The rate constant for the one-compartment reactor was 1.3 times higher than the two-compartment reactor ($k = 0.3724 \, h^{-1}$ vs. $k = 0.2817 \, h^{-1}$); thus, the improvement is not major and the separation allows the use of different electrolytes in each compartment and can enable the production of secondary product, e.g., $H_2$ or $H_2O_2$ [65,97,98].

There is also the ability for dual water treatment, and Byrne et al. [99] examined the use of a two-compartment cell separated by a glass frit, using a $TiO_2$ photoanode prepared by electrophoretic deposition of P25 powder and a copper wire counter electrode for the oxidation of organic pollutants (oxalate) and recovery of metal ions, i.e., copper reduction. The anodic compartment contained 15 mM of potassium oxalate in KCl (pH = 6.5) and the cathodic compartment cell contained 1.25 mM $CuSO_4$ (pH = 1.5). The rector was irradiated with a 150 W Xe lamp and removed >94.4% of the copper in the cathodic solution and there was a 9% reduction in TOC for the removal of oxalate after 105 min. Byrne et al., subsequently reported the use of a two-compartment flat plate sandwich type reactor for the simultaneous oxidation of oxalate and the recovery of copper [72]. The photoanode was P25 $TiO_2$ coated on FTO glass under back-face irradiation, with an anion exchange membrane separating the anode from a copper mesh counter electrode. The irradiation source was two Philips TLD 18 W/08 UVA lamps. The Faradaic yield for formic acid degradation was 126% which was explained by photocurrent losses due to residual $O_2$ at the anode. The Faradaic yield for copper recovery was 95.5%.

Although this review is based on aqueous PEC water treatment, PEC gaseous/air treatment has also been investigated [100]. A brief example is the two-compartment design of Mei et al. [101] tested with an AM 1.5 (100 mWcm$^{-2}$) source for the oxidation of sulphur dioxide ($SO_2$) to sulphate ($SO_4^{-2}$) which requires a strong alkaline electrolyte (NaOH was used as the anolyte in this study, pH = 13.65), with a nanorod-like hematite photoanode, which is reported to have high durability and corrosion resistance in alkali environments [102]. The counter electrode was a gas diffusion electrode to produce $H_2O_2$,

the catholyte was 0.1 M $Na_2SO_4$ in $H_2SO_4$ (pH = 3) and the electrolytes were separated by a bipolar membrane. The addition of $SO_2$ also increased the photocurrent to 1.2 $mAcm^{-2}$, leading to improved $H_2O_2$ concentrations (58.8 $\mu molL^{-1}h^{-1}cm^{-2}$). Therefore, the system facilitates the synergistic treatment of desulphurisation of flue gas and simultaneous production of high-value $H_2O_2$, of which the price per mole is 1.5 times the value of $H_2$.

### 5.10. Hybrid System

In a hybrid system, there are two separate solutions/cells, and unlike the two-compartment reactor there is no ionic transport between the solutions; instead, the electrons are transported via an external circuit, as shown in Figure 24. One of the cells can act as a photocatalytic fuel cell to generate a potential bias that can power the separate PEC cell. In the design of Liu et al. [103], the fuel cell used a $TiO_2$ nanotube array as the photoanode and a Pt-black/Pt counter in a solution of 0.05 M acetic acid and 0.1 M $Na_2SO_4$ which generated a 0.608 V open-circuit voltage and a short-circuit current of 0.261 mA. The PEC cell used a nanotube array as a photoanode and a Pt counter for the degradation of 0.045 mM tetracycline. The inclusion of the fuel cell increased the rate constant from 0.318 $h^{-1}$ photocatalytically, i.e., without the fuel cell, to 0.555 $h^{-1}$. The configuration of Nordin et al. [104] examined the use of three different photocatalysts ($TiO_2$, ZnO, $\alpha$-$Fe_2O_3$) immobilised on carbon cloth for the fuel cell photoelectrode, of which ZnO generated the highest open-circuit potential of 82.3 mV and a photocurrent of ~0.085 mA. The second cell used a carbon plate anode and an iron plate cathode, forming a peroxi-coagulation cell. The reactor was tested using 10 $mgL^{-1}$ of amaranth dye in both cells, and the fuel cell removed 83.3%, 93.8% and 4.0% and the peroxi-coagulation cell 91.4%, 86.9% and 87.5% for $TiO_2$, ZnO and $\alpha$-$Fe_2O_3$, respectively.

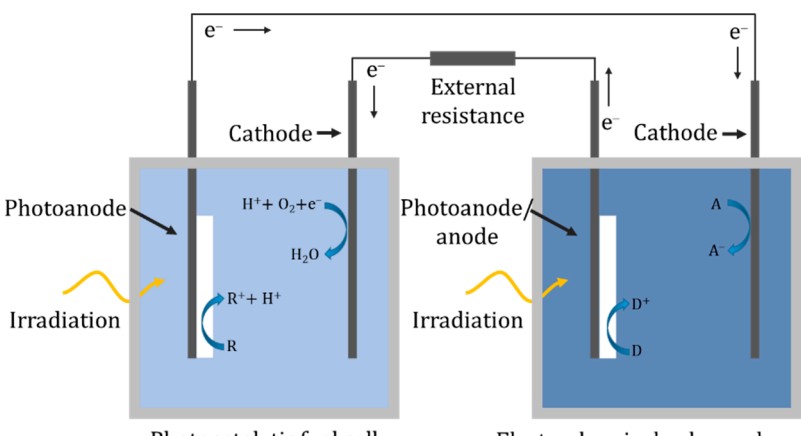

**Figure 24.** Hybrid system with a fuel cell used to self-bias the system and a secondary compartment for additional reactions.

### 5.11. Three-Dimensional Electrode Reactor

The final and somewhat derivative category of PECs is the use of a three-dimensional electrode. The anode and cathode in this instance are traditional electrochemical electrodes. The 3D electrode is the photocatalyst and is used as a secondary method to enhance the performance by utilising photocatalysis and potentially by acting as a bipolar electrode. The 3D electrode can be in the form of a slurry reactor [105], immobilised onto millimeter-scale substrates to form a packed bed reactor [106] or as a singular porous electrode [107].

The slurry design of An et al. [105], shown in Figure 25, used a stainless steel anode and cathode, with granulated activated carbon and $TiO_2$ as the 3D electrode for the degradation of reactive brilliant red X-3B. The reactor was aerated from the bottom with the use of a micropore plate to provide oxygen and mixing and keep the particulates in suspension. The addition of the 3D electrode improved the kinetic rate ($k$ = 0.1067 $min^{-1}$) compared to only photocatalytic degradation ($k$ = 0.0351 $min^{-1}$) or electrochemical oxidation

($k$ = 0.0562 min$^{-1}$). Post-treatment filtration is still required to separate the photocatalyst from the solution, and cost/energy analysis also needs to be performed to examine if the additional complexity and input energy are worth the increased efficiency.

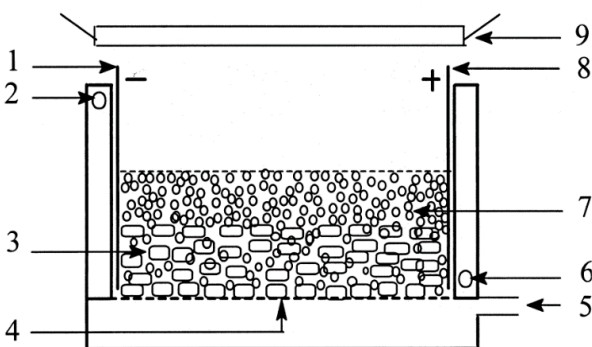

**Figure 25.** Schematic diagram of the three-dimensional electrode slurry photocatalytic reactor. (1) Cathode; (2) outlet of recycled water; (3) GAC layer; (4) micropore plate; (5) inlet of pressured air; (6) inlet of recycled water; (7) titanium dioxide layer; (8) anode; (9) UV irradiation. (Reproduced with permission from [105]: An, T.; Zhu, X.; Xiong, Y. Synergic degradation of reactive brilliant red x-3b using three dimension electrode-photocatalytic reactor. *J. Environ. Sci. Health A* **2001**, *36*, 2069–2082, doi:10.1081/ESE-100107449. Copyright (2001), Taylor & Francis (Abingdon, UK)).

To remove the need for filtration, the photocatalyst can be immobilised onto millimeter-scale substrates, i.e., beads or Raschig rings [108], to be used as the 3D electrode, as in the design of Liu et al. [106] shown in Figure 26.

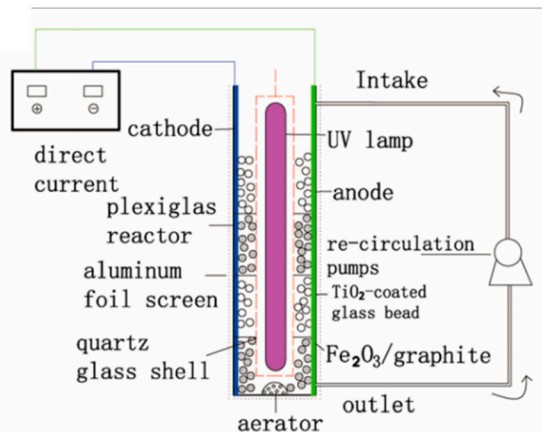

**Figure 26.** Schematic diagram of the three-dimensional electrode packed bed photocatalytic reactor. (Reproduced with permission from [106]: Liu, R.; Zhang, Y. Highly efficient degradation of berberine chloride form wastewater by a novel three-dimensional electrode photoelectrocatalytic system. *Environ. Sci. Pollut. Res.* **2018**, *25*, 9873–9886, doi:10.1007/s11356-018-1293-y. Copyright (2018), Springer Nature (Heidelberg, Germany)).

Their system used $Fe_2O_3$/graphite and $TiO_2$-coated glass beads as the 3D electrode. The anode and cathode were titanium and stainless steel, respectively, operated with a current density of 9 mAcm$^{-2}$ and with a 40 W UV lamp (78 µWcm$^{-2}$). The inclusion of the 3D electrode improved the rate of berberine chloride form degradation, and a rate constant ($k$) of 0.015 min$^{-1}$ was achieved with the 3D electrode under irradiation, compared to $k$ = 0.007 min$^{-1}$ without irradiation and $k$ = 0.001 min$^{-1}$ without irradiation and the 3D electrode. The efficiency analysis shows that the 3D electrode had a higher mineralisation current efficiency, monitored by TOC and using Equation (7) [109], where n is the number of electrons consumed during mineralisation, F is the Faraday constant (96,487 C mol$^{-1}$),

V is the volume (L), $\Delta$TOC is the amount of TOC removed (mg L$^{-1}$), $4.32 \times 10^7$ is a conversion factor (3600 s h$^{-1}$ $\times$ 12,000 C mol$^{-1}$), m is the number of carbon atoms in the compound, I is the current (A) and t is the time elapsed (h). The 3D electrode also had a lower energy consumption per unit of TOC mass and therefore, by extension, a lower energy cost calculated by Equation (8), where Energy$_{(TOC)}$ is the energy required for a unit of TOC (kWh mg$^{-1}$$_{(TOC)}$), P is the total electrical power of all used appliances (kW), t is the time elapsed (h) and $\Delta$(TOC) is the amount of TOC removed (mg). Alternatively, the energy required can be expressed as per volume for a set amount degradation/order of efficiency (kWh m$^{-3}$) using Equation (9), where V is the volume (m$^3$) [110].

$$\text{Mineralisation current efficiency} = \frac{n\,F\,V\,\Delta(TOC)}{4.32 \times 10^7 m\,I\,t} \tag{7}$$

$$\text{Energy}_{(TOC)} = \frac{P\,t}{\Delta(TOC)} \tag{8}$$

$$\text{Energy}_{(L)} = \frac{P\,t}{V} \tag{9}$$

The 3D electrode can also take the form of a singular porous bipolar electrode, as with the design of Mesones et al. [107], in which the 3D electrode is a composite of granular activated carbon and TiO$_2$ contained within a basket (Figure 27), and a dimensionally stable anode (DSA) made of RuOx/Ti and a stainless steel cathode were used, for the inactivation of *E. coli* K12 in a high-salt solution (NaCl 35 g L$^{-1}$) similar to fish farm seawater. The results demonstrated that there was no synergic effect when combining the photocatalytic process with the electrochemical process. The electrochemical set-up which produced chlorine radicals had the lowest energy requirement of 0.007 kWh m$^{-3}$ at 0.1 mA cm$^{-2}$; the inclusion of the lamp and 3D electrode increased the energy requirement to 2.2 kWh m$^{-3}$.

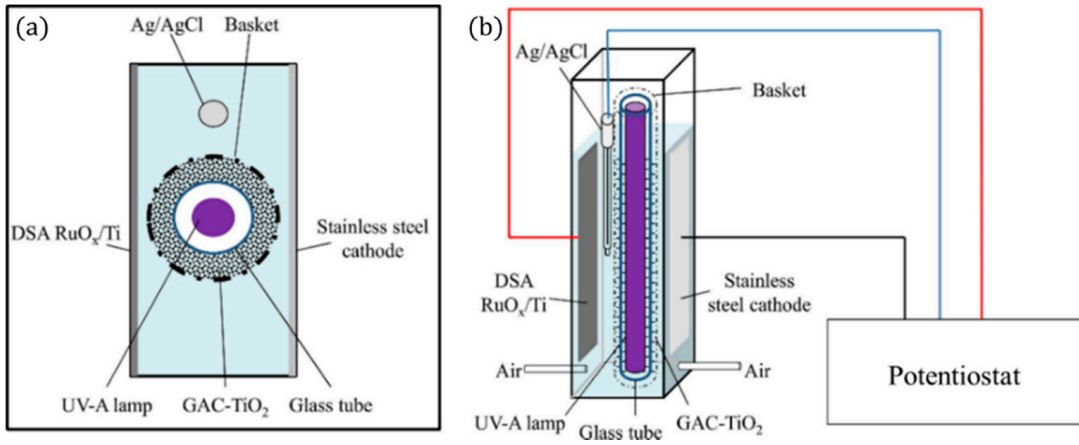

**Figure 27.** Schematic representation of the 3D PEC reactor, (**a**) upper view and (**b**) front view. (Adapted with permission from [107]: Mesones, S.; Mena, E.; López-Muñoz, M.J.; Adán, C.; Marugán, J. Synergistic and antagonistic effects in the photoelectrocatalytic disinfection of water with TiO$_2$ supported on activated carbon as a bipolar electrode in a novel 3D photoelectrochemical reactor. *Sep. Purif. Technol.* **2020**, *247*, 117002, doi:10.1016/j.seppur.2020.117002. Copyright (2020), Elsevier (Amsterdam, The Netherlands)).

## 6. Conclusions and Prospects

Photoelectrocatalysis can improve the performance of immobilised photocatalytic systems by reducing recombination, producing secondary ROS at the counter electrode and enabling electromigration of charge species towards the electrodes. Photoelectrodes are a critical component in a PEC system and can underpin the overall performance. It is difficult, however, to compare the results reported for different photoelectrodes due to experimental variations, such as reactor set-up, irradiation, geometry, target contaminants

and operational conditions, i.e., potential and concentration and type of electrolyte. Lower bandgap materials should offer improved solar performance by utilising more visible light, and there have been extensive investigations, but these have been primarily focused on solar fuels (water splitting and $CO_2$ reduction) research rather than water treatment.

Maximising the effectiveness of any adequate photoelectrode reactor design is required. This review identified eleven distinct PEC reactor design categories; however, comparing reactor designs to find the optimal one is difficult as each type has its advantages and disadvantages; selection or the improvement of an existing design should be based on the application, associated cost and potential to scale it up. Despite this, several conclusions can be drawn from the literature examined that impact any reactor design. A high surface to volume ratio with minimal distance between electrodes is favourable. Mass transport can impact/improve the degradation rates. An adequate supply of oxygen is important for the reductive pathways to produce ROS. Depending on the contaminant and electrolyte used, there is an optimal bias. A range of different counter electrodes (stainless steel, carbon cloth, carbon paper, platinum, titanium) have been used in different reactor set-ups but selection/optimisation of the counter electrode is rarely examined for its incorporation with PECs.

Few studies have been conducted using real water sources and, given the complex nature of real water, this would be expected to result in lower degradation rates due to competing reactions such as the oxidation of natural organic matter or carbonates. The use of standard electrolyte solutions can increase the conductivity optimal or suboptimal values, and can introduce beneficial ions such as chloride in concentrations higher than what would be present in reality, but ensures a standardised method of testing at the laboratory scale. Further testing and analysis using a real water matrix is therefore required after initial testing. An examination of the water matrix and its influence on the overall treatment for specific applications is also required; for example, examining the ions/compounds present, salinity, organic matter and turbidity. Limited testing has been performed under real sunlight. The use of artificial irradiation may enable the construction of continually operational PEC reactors if the irradiation source has acceptable energy efficiency (it may be powered by a renewable source) and the system is cost-effective.

As most of the EAP research reported is still at lab or bench scale, there are no reports on the techno-economic and scaling up of these systems. More research on the optimisation of the photoelectrode, counter electrode, reactor design, operation and modelling is still required to enhance the process and move towards pilot-scale testing. It is unlikely that one technology will solve all the challenges in water and wastewater treatment but PECs has the potential to be used in niche applications or as part of multi-stage treatment systems.

**Author Contributions:** Writing—Original Draft, Visualisation, Writing—Review and Editing (S.M.); Supervision, Writing—Review and Editing, Funding Acquisition (P.F.-I.); Supervision, Writing—Review and Editing, Funding Acquisition (J.A.B.). All authors have read and agreed to the published version of the manuscript.

**Funding:** Department for Economy (DfE) Northern Ireland for funding Stuart McMichael. SAFEWATER project sponsored by Global Challenges Research Fund (GCRF) UK Research and Innovation (SAFEWATER; EPSRC Grant Reference EP/P032427/1).

**Institutional Review Board Statement:** Not applicable.

**Informed Consent Statement:** Not applicable.

**Data Availability Statement:** Not applicable.

**Conflicts of Interest:** The authors declare no conflict of interest.

## Appendix A

**Table A1.** Selection of PEC reactors, their operational parameters, and selected results. DL—detection limit, % amount of reduction.

| Reactor Type | Photoanode | Counter Electrode | Irradiation Type | Electrolyte | Volume or Flow Rate | Electrical Mode of Operation | Target Contaminant(s) | Selected Results/Findings | Year of Pub. | Ref. |
|---|---|---|---|---|---|---|---|---|---|---|
| Experimental | $N_2$-doped anodised titanium nanotubes | Pt paddle | 450 W Xe | 1/4 strength Ringer's solution | 0.035 L | +1.0 V (SCE) | *E. coli* $10^6$ CFU mL$^{-1}$ | NT 120 min—DL Nitrogen NT 60 min—DL | 2017 | [54] |
| Experimental | $TiO_2$ film on ITO | Stainless steel propeller | $2 \times 9$ W UVA 370 nm $2 \times 9$ W UVB 310 nm | No additional supporting electrolyte | 0.2 L | 0 V to +3.0 V (SCE) | Formic acid 5.3 mM | Optimal results +10 V (SCE) | 2005 | [56] |
| Experimental | ALD of $Fe_2O_3$ onto $Bi_2WO_6$ on FTO | Pt wire | 150 W Xe AM 1.5 filter 100 mWcm$^{-2}$ | 0.5 M $Na_2SO_4$ | 0.03 L | +0.6 V (Ag/AgCl) | Tetracycline 20 mg L$^{-1}$ | The electrode was used for both degradation and detection 95% degradation after 90 min | 2019 | [52] |
| Cylindrical | $TiO_2$ immobilised on ITO glass | Nickle mesh 55% open area | 6 W black light lamp 362 nm Peck 0.047 mWcm$^{-2}$ | 0.1 M $Na_2SO_4$ pH = 6.4 | 1 L | Cell bias 0 to +1.4 V | *E. coli* $10^3$ CFU mL$^{-1}$ | PC—300 min DL 0.4 V—280 min DL 1 V—~220 min DL 1.4 V—140 min DL | 2017 | [46] |
| Cylindrical | Anodised titanium nanotube cylinder sheet | DSA De Nora | 36 W UVB Lamp | 0.01 M $Na_2SO_4$ | 1 L | Cell Bias +1.0, +1.5, +2.0 V | Benzophenone-3 10 mg L$^{-1}$ *Candida parapsilosis* $10^6$ CFU mL$^{-1}$ | TOC removal (20 min) $O_3$ + EAP 2 V > EAP 2 V > $O_3$ 60 min for 6-log reduction | 2019 | [47] |

Table A1. *Cont.*

| Reactor Type | Photoanode | Counter Electrode | Irradiation Type | Electrolyte | Volume or Flow Rate | Electrical Mode of Operation | Target Contaminant(s) | Selected Results/Findings | Year of Pub. | Ref. |
|---|---|---|---|---|---|---|---|---|---|---|
| Cylindrical | Anodised titanium nanotube cylinder mesh | Activated titanium electrode De Nora | 15 W 254 nm Lamp | KCl $635 \pm 15$ $\mu Scm^{-1}$ | 1.8 L | 0, +1.0, +1.5 V (Ag/AgCl) | Reactive Red 243 25 mg $L^{-1}$ | 90% decolourisation in 45 min 99% decolourisation in 60 min | 2018 | [111] |
| Cylindrical | Anodised $TiO_2$ nanotubes | Titanium | 14 W UV lamp 275 nm | 0.02 M $Na_2SO_4$ | 1 L total 180 mL $min^{-1}$ | Cell bias +8 V | Acetaminophen 10 mg $L^{-1}$ | EAP 95% at 8 V EE 3% at 8 V PC 72% | 2019 | [63] |
| Flat reactor | Anodised titanium nanotubes | Pt sheet | UV lamp 254 nm peak 5/10 $mWcm^{-2}$ | 0.1 M $Na_2SO_4$ | 0.05 L | Cell Bias +1.0 V | Tetracycline 100–400 mg $L^{-1}$ | 90% reduction 100 mg—5 $mWcm^{-2}$—47 min 100 mg—10 $mWcm^{-2}$—16 min 400 mg—10 $mWcm^{-2}$—52 min | 2010 | [71] |
| Flat reactor | $TiO_2$ immobilised on FTO glass | Carbon cloth | 6 W 365 nm Lamp | $K_2SO_4$ 0.05 M pH = 3 | 0.2 L 0.1 L $min^{-1}$ flow rate | 0.62–2.5 $mAcm^{-2}$ | 1.4 mM phenol | Improved degradation rates when air pumped into reactor 0.0127 $min^{-1}$ vs. 0.0051 $min^{-1}$ | 2017 | [68] |
| Flat reactor | Anodised $TiO_2$ nanotubes | Carbon paper/platinum mesh | Two 9 W UVA Lamp 5 $mWcm^{-2}$ | Surface water 697 $\mu Scm^{-1}$ | 0.19 L | Cell bias +1.0 V | *E. coli* $10^6$ CFU $mL^{-1}$ | Carbon paper—2.7-log reduction Platinum—2.0-log reduction | 2021 | [112] |

Table A1. *Cont.*

| Reactor Type | Photoanode | Counter Electrode | Irradiation Type | Electrolyte | Volume or Flow Rate | Electrical Mode of Operation | Target Contaminant(s) | Selected Results/Findings | Year of Pub. | Ref. |
|---|---|---|---|---|---|---|---|---|---|---|
| Channel reactor | $BiVO_4$ on ITO | ITO glass | Blue LED at 10 V 402 nm 80 mWcm$^{-2}$ | 0.1 M NaCl | 75 uL min$^{-1}$ | Cell bias −2.2 V to +2.2 V | Methylene blue 0.03 mM | Optimal results with positive +1.5 V k = 0.064 s$^{-1}$ Optimal results with negative −2.2 V k = 0.102 s$^{-1}$ | 2012 | [49] |
| Channel reactor | Anodised titanium nanotubes | Pt | UV LED 365 nm 8 mWcm$^{-2}$ | 0.1 M NaNO$_3$ | 19.25 uL min$^{-1}$ | +0.7 V (Ag/AgCl) | *E. coli* 10$^7$ CFU mL$^{-1}$ | NT better than particulate film, >6-log reduction 97 s | 2013 | [48] |
| Channel reactor | H$_2$ annealed titanium anodised nanotubes | Stainless steel | Real solar average total 100 mWcm$^{-2}$ | Natural lake water 50 mg L$^{-1}$ NaCl added | 20/50 mL min$^{-1}$ | +6.0 V | *E. coli* 165 CFU 100 mL$^{-1}$ | >7 V NT delaminate *E. coli* CFU/100 mL +6.0 V 20 mL min$^{-1}$ = 38 (light) +6.0 V 20 mL min$^{-1}$ = 50 (dark) +6.0 V 50 mL min$^{-1}$ = 0 (light) +6.0 V 50 mL min$^{-1}$ = 0 (dark) | 2016 | [70] |
| Rotational | Anodised titanium nanotubes on sheet | Ti sheet | 1 kW Xe lamp 400–300 nm 5.4 mWcm$^{-2}$ | NaCl 1.03 mScm$^{-1}$ | 0.2 L | Ag/AgCl +0.5 to +3.5 V | Methylene blue 2–5 ppm | 180 min +1.5 V 86% +2.5 V 90.4% | 2017 | [78] |
| Rotational | Dip coating TiO$_2$ on Ti | Cu sheet | 11 W lamp 254 nm 15 mWcm$^{-2}$ | 3.5 mM Na$_2$SO$_4$ | 0.055 L | Cell bias +0.4 V | Rhodamine B 20 mg L$^{-1}$ | 1 h 82% 20 mg L$^{-1}$ | 2008 | [26] |

Table A1. *Cont.*

| Reactor Type | Photoanode | Counter Electrode | Irradiation Type | Electrolyte | Volume or Flow Rate | Electrical Mode of Operation | Target Contaminant(s) | Selected Results/Findings | Year of Pub. | Ref. |
|---|---|---|---|---|---|---|---|---|---|---|
| Fibre optic | Modified $SiO_2$ fibre with $SnO_2$:Sb and $TiO_2$ film | Carbon cloth | UV mercury lamp 254 nm 2.1 mWcm$^{-2}$ | 0.05 M $Na_2SO_4$ pH = 3 using $H_2SO_4$ | 0.64 L 80 L h$^{-1}$ | +1.0 mAcm$^{-2}$ | Orange II 15 mg L$^{-1}$ | Electro k = 0.0122 min$^{-1}$ EAP k = 0.0126 min$^{-1}$ Electro-Fenton k = 0.1956 min$^{-1}$ Electro-Fenton EAP k = 0.2303 min$^{-1}$ | 2009 | [85] |
| Membrane | g-$C_3N_4$/CNTs/$Al_2O_3$ membrane | Titanium mesh | 300 W Xe lamp 100 mWcm$^{-2}$ | 0.01 M $Na_2SO_4$ | 1.25 mL min$^{-1}$ | Cell bias 0.0 to +1.5 V | Phenol 5 mg L$^{-1}$ | PC 26% EE +1.0 V 36% EAP +1.0 V 71% EE +1.5 V 81% EAP +1.5 V 94% | 2017 | [89] |
| Membrane | Nylon/stainless steel-$WO_3$ | Platinum network | Xe 300 W | 0.1 M $Na_2SO_4$ pH = 6 | 50 mL/100 mL min$^{-1}$ | +1.0 V (Ag/AgCl) | RR-120 0.01 μM | After 90 min Photolysis 0% PC~35% EAP~50% | 2020 | [92] |
| Membrane | Polyvinylidene fluoride on carbon felt and $ZnIn_2S_4$ microparticles | Titanium plate | Xe 300 W | 0.05 M $Na_2SO_4$ pH = 6.5 | 2 L | Cell bias −0.9 to +0.9 V | Tetracycline 4 mg L$^{-1}$ | Optimal bias occurred at −0.3 V pH = 6.5 87% after 180 min | 2020 | [94] |
| Two-compartment | $TiO_2$ on Ti by dip coating | Silver sheet | 4 × 6 W UV lamp 365 nm | 0.5 M $Na_2SO_4$ | 0.1 L | 0.0, +0.5, +1.0, +1.5, +2.0 (SCE) | Aniline 10 mg L$^{-1}$ | The single-compartment reactor had higher degradation than the two-compartment reactor | 2003 | [96] |

Table A1. *Cont.*

| Reactor Type | | Photoanode | Counter Electrode | Irradiation Type | Electrolyte | Volume or Flow Rate | Electrical Mode of Operation | Target Contaminant(s) | Selected Results/Findings | Year of Pub. | Ref. |
|---|---|---|---|---|---|---|---|---|---|---|---|
| Two-compartment | | Sol–gel $TiO_2$ on Ti | Platinum spiral | 300 W UV lamp 365 nm | 0.5 M $Na_2SO_4$ | 0.1 L | 0.0, +0.5 V (SCE) | Aniline 10 mg $L^{-1}$ Salicylate 0.5 mM | Dual degradation and $H_2O_2$ production in the cathodic compartment | 2006 | [98] |
| Two-compartment | | 1D $\alpha$-$Fe_2O_3$ nanorods on FTO | Gas diffusion electrode | AM 1.5 100 $mWcm^{-2}$ | Anolyte NaOH pH = 13.65 Catholyte 0.1 M $Na_2SO_4$ in $H_2SO_4$ pH = 3 | 0.04 L | +0.5 V (Ag/AgCl) | $SO_4^{-2}$ | $H_2O_2$ production rate 58.8 µmol $L^{-1}h^{-1}cm^{-2}$. The addition of $SO_2$ increased the photocurrent | 2020 | [101] |
| Hybrid | Fuel cell | $TiO_2$ nanotubes on Ti foil | Pt-black/Pt | UV 2.0 $mWcm^{-2}$ | 0.05 M acetic acid and 0.1 M $Na_2SO_4$ | 0.01 L | Self-biased by fuel cell +0.550 V, current 0.086 mA | - | The EAP 0.555 $h^{-1}$, PC 0.318 $h^{-1}$ (no fuel cell) Photolysis 0.057 $h^{-1}$ Electrochemical 0.003 $h^{-1}$ | 2012 | [103] |
| | PEC | $TiO_2$ nanotubes on Ti foil | Pt | | 0.1 M $Na_2SO_4$ | 0.025 L | | Tetracycline 0.045 mM | | | |
| Hybrid | Fuel cell | ZnO on carbon cloth | Carbon plate | UVA 36 W | Not stated | 0.5 L | Self-biased by fuel cell 82.3 mV | Amaranth dye 10 $mgL^{-1}$ | 93.8% in the fuel cell 86.9% removal in the EE | 2019 | [104] |
| | EE | Carbon plate | Iron plate | - | $H_2SO_4$ to adjusted pH to 3 | 0.5 L | | | | | |
| CPC/Two-compartment | | Dip coating $TiO_2$ on Ti | Pt/$SnO_2$ glass substrate | Real solar | Working 0.02 M $H_2SO_4$ Inner 0.2 M $H_2SO_4$ | 4.5 L | +1.4 V (RHE) | 4Cl-phenol and pyrimethanil 20–30 mg $L^{-1}$ | $TiO_2$ slurry: 0.106 mg $min^{-1}$ $m^{-2}$ 4Cl-phenol: 7.3 mg $min^{-1}$ $m^{-2}$ Pyrimethanil: 6.21 mg $min^{-1}$ $m^{-2}$ | 1999 | [65] |

Table A1. *Cont.*

| Reactor Type | Photoanode | Counter Electrode | Irradiation Type | Electrolyte | Volume or Flow Rate | Electrical Mode of Operation | Target Contaminant(s) | Selected Results/Findings | Year of Pub. | Ref. |
|---|---|---|---|---|---|---|---|---|---|---|
| CPC | Anodised titanium nanotubes | Carbon felt | Real Solar | Harvested rainwater 70 $\mu$Scm$^{-1}$ | 0.3 L | +1.0 V cell | *E. coli* *P. aeruginosa* Both >7 Log | 5.5-log reduction *E. coli* 5.8-log reduction *P. aeruginosa* EMA-qPCR used for molecular viability analysis | 2021 | [67] |
| Three-dimensional electrode | GAC-TiO$_2$ bipolar photoelectrode (DSA RuOx/Ti anode) | Stainless steel sheet | 6 W 365 nm lamp | Synthetic seawater salt concentration of 35 g L$^{-1}$ pH = 6.2 | 0.55 L | 0.03, 0.06 & 0.1 mAcm$^{-2}$ | *E. coli* $10^3$ CFU mL$^{-1}$ | Disinfection results are similar to the non-electrochemical process. The high energy usage of the lamp decreases the efficiency of the 3D electrode | 2020 | [107] |
| Three-dimensional electrode | GAC and TiO$_2$ slurry (stainless steel anode) | Stainless steel | Hg 500 W 365 nm 6.64 mWcm$^{-2}$ | No supporting electrolyte stated | 0.2 L | 0 to +30.0 V | Brilliant red X-3B 1.0 mM | Removal in 30 min, the slurry performs better than the electric-only with lower potentials | 2001 | [105] |
| Three-dimensional electrode | Fe$_2$O$_3$/graphite and TiO$_2$-coated glass beads (titanium anode) | Stainless steel | 40 W UV 0.078 mWcm$^{-2}$ | 0.1 M Na$_2$SO$_4$ pH = 3 optimal | 0.6 L | 9.0 mAcm$^{-2}$ | Berberine chloride form 200 mg L$^{-1}$ | 93% removal after 60 min Enhanced performance with a 3D electrode and lower energy cost | 2018 | [106] |

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
