# Peer review of "A Review of Photoelectrocatalytic Reactors for Water and Wastewater Treatment"

_water, doi:10.3390/w13091198_

Round 1

Reviewer 1 Report

I appreciated this review, considering the interest towards these types of processes.

I recommend the publication.

In my opinion a careful check of the manuscript for punctuations, typos, and some grammar structures should be useful: I’m not an english mother-tongue but some sentences are difficult to read.

Why do authors refer only to UVA lamps (line 250), neglecting UV lamp emitting in other wavelength range (as the UVC)?

Considering the fundamental role of supporting electrolyte (also simply measured as conductivity) in PEC performing, I would like to find a comment on the applicability of these processes to real water matrices depending on their salinity. This aspect could greatly affect the application, and providing some insights would be useful for researchers interested in this topic.

Author Response

We would like to thank the reviewer for their feedback:

Potint-by-point response: 

In my opinion a careful check of the manuscript for punctuations, typos, and some grammar structures should be useful: I’m not an english mother-tongue but some sentences are difficult to read.

  • The manuscript has been reviewed, typos, grammar and structure changed were required to improve the clarity for the reader

Why do authors refer only to UVA lamps (line 250), neglecting UV lamp emitting in other wavelength range (as the UVC)?

  • Wording changed to "UV fluorescent lamps". Thus the including the full UV spectrum.  

Considering the fundamental role of supporting electrolyte (also simply measured as conductivity) in PEC performing, I would like to find a comment on the applicability of these processes to real water matrices depending on their salinity. This aspect could greatly affect the application, and providing some insights would be useful for researchers interested in this topic.

  • The role of the supporting electrolyte and overall water matrix is extremely important in PEC and the overall performance. Only a few papers examine the use of real water solutions, the information that was available from the respective papers is sometimes limited on the analyse of the ion composition/salinity, but we have included what was available. We have throughout the document highlight that Cl- can be beneficial to the PEC process and reported for example the reactor design by Mesones et al. line 762 in which they used simulated water (NaCl 35g/L) similar to fish farm seawater. Highlighted that the electrolyte solution (turbidity) can affect if usage of front-face / back face (line 243).  The salinity/water matrix will greatly depend and change on the target water source (e.g. wastewater and stream water), a real water matrix can often be rather complex. We have also added in the conclusions and prospects that it is an area that requires attention and examination.

Reviewer 2 Report

The review paper reports a concise and accurate summary of the latest studies on photoelectrocatalytic reactors for water and water treatments. The manuscript is very clear and well-structured despite the complexity of the topic.

Therefore, the manuscript could be accepted for publication in MDPI.

Just a few suggestions are given below:

- In the abstract, I suppose that the words “photoelectrodes” (lines 16-17) and “key finding finds” are typos. Please check.

- In line 48, “inactivate micrograms” should be replaced with “inactivate microorganisms”.

-  In figure 1, the correct form should be “hν ≥ Eg” instead of “Eg ≥ hν”. Please check it.

- When the Authors explain the photocatalytic mechanism, some references should be also added. 

- It is recommended to specify the meaning of “D” (donor) the first time it is reported.

Author Response

We would like to thank the reviewer for the fast & positive feedback, as well as the helpful suggestions.  

Point-by-point response:

- In the abstract, I suppose that the words “photoelectrodes” (lines 16-17) and “key finding finds” are typos. Please check. 

  • Typos fixed 

- In line 48, “inactivate micrograms” should be replaced with “inactivate microorganisms”.

  • Spelling corrected

-  In figure 1, the correct form should be “hν ≥ Eg” instead of “Eg ≥ hν”. Please check it.

  • Diagram updated to the correct "hν ≥ Eg"

- When the Authors explain the photocatalytic mechanism, some references should be also added. 

  • References added 

- It is recommended to specify the meaning of “D” (donor) the first time it is reported.

  • Doner has been added when "D" is used the first time 

Reviewer 3 Report

Dear authors of the entitled manuscript 'A review of photoelectrocatalytic reactors for water and wastewater treatment', after the revision of your work the referee can´t find any aspect or comment to correct or improve your manuscript.

The paper is well organized and all the sections are properly explained. The quality of the figures is very high and the tables provide detailed information. 

The referee only can congratulate the authors for their sensational work and hopes that it will be highly cited.

Author Response

We would like to thank the reviewer for their fast and positive feedback.